# ExCeL: Combined Extreme and Collective Logit Information for Out-of-Distribution Detection

**Naveen Karunanayake**                                    *naveen.karunanayake@sydney.edu.au*
*School of Computer Science*
*The University of Sydney*

**Suranga Seneviratne**                                    *suranga.seneviratne@sydney.edu.au*
*School of Computer Science*
*The University of Sydney*

**Sanjay Chawla**                                          *schawla@hbku.edu.qa*
*Qatar Computing Research Institute, HBKU*

**Reviewed on OpenReview:** *https://openreview.net/forum?id=4Xz0WBAiX4*

## Abstract

Deep learning models often exhibit overconfidence in predicting out-of-distribution (OOD) data, underscoring the crucial role of OOD detection in ensuring reliability in predictions. Among various OOD detection approaches, post-hoc detectors have gained significant popularity, primarily due to their ease of implementation and competitive performance. However, recent benchmarks for OOD detection have revealed a lack of consistency in existing post-hoc methods. This inconsistency in post-hoc detectors can be attributed to their sole reliance either on *extreme information*, such as the maximum logit, or on *collective information* (i.e., information spanned across classes or training samples) embedded within the output layer. In this paper, we propose ExCeL, which combines both extreme and collective information within the output layer for enhanced and consistent performance in OOD detection. We leverage the logit of the top predicted class as the extreme information (i.e., the maximum logit), while the collective information is derived in a novel approach that involves assessing the probability of other classes appearing in subsequent ranks across various training samples. Our idea is motivated by the observation that, for in-distribution (ID) data, the ranking of classes beyond the predicted class is more deterministic compared to that in OOD data. Experiments conducted on CIFAR100, ImageNet-200, and ImageNet-1K datasets demonstrate that ExCeL consistently is among the five top-performing methods out of twenty-one existing post-hoc baselines when the joint performance on near-OOD and far-OOD is considered (i.e., in terms of AUROC and FPR95). Furthermore, ExCeL shows the best overall performance across all datasets, unlike other baselines that work best on one dataset but have a performance drop in others.

## 1 Introduction

Deep neural networks (DNNs) deployed in the open world often encounter a diverse range of inputs from unknown classes, commonly referred to as out-of-distribution (OOD) data. However, DNNs' tendency to be overly confident, yet inaccurate about such inputs makes them less reliable, particularly in safety-critical applications such as autonomous driving (Filos et al., 2020) and healthcare (Roy et al., 2022). Therefore, a DNN should be able to identify and avoid making predictions on these OOD inputs that differ from its training data. For instance, in an autonomous vehicle, the driving system must promptly alert the driver and transfer control when it detects unfamiliar scenes or objects that were not encountered during its

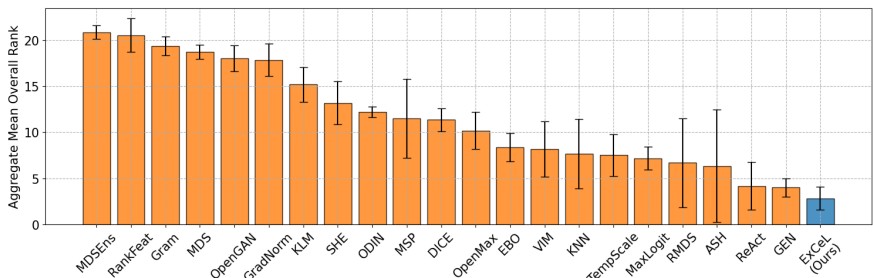

Figure 1: The performance comparison of 22 post-hoc OOD detection algorithms based on *aggregate mean overall rank* (i.e., mean overall rank averaged across all datasets). **ExCeL** exhibits the highest consistency across all datasets, indicated by its lowest aggregate mean overall rank (**cf.** Section 6.1).

training (Nitsch et al., 2021). Accordingly, addressing the challenge of OOD detection has gained significant attention in recent studies (Yang et al., 2021b).

Among various OOD detection methods, post-hoc inference techniques (Hendrycks & Gimpel, 2016; Liang et al., 2017; Lee et al., 2018; Liu et al., 2020; Sun et al., 2021; Hendrycks et al., 2019a; Wang et al., 2022; Sun et al., 2022; Sun & Li, 2022) stand out due to their ease of implementation and competitive performance (Yang et al., 2021b; Zhang et al., 2023b). They can be applied to any pre-trained model, making them versatile and applicable without the need for modifications during the training phase. These techniques extract crucial information from intermediate (Sastry & Oore, 2020; Huang et al., 2021) or output (Hendrycks & Gimpel, 2016; Hendrycks et al., 2019a; Liu et al., 2020; Lee et al., 2018) layers of DNNs, establishing an OOD score to distinguish between in-distribution (ID) and OOD samples. Since the output layer (i.e., normalised or unnormalised probabilities) in a DNN adeptly captures high-level semantics (i.e., objects, scenes etc.), researchers have increasingly focused on employing its features for OOD detection. For example, Hendrycks & Gimpel (2016) proposed the *maximum* softmax probability (MSP) as an initial baseline for OOD detection. Subsequently, Hendrycks et al. (2019a) and Liu et al. (2020) further harnessed more *extreme* information from the output layer for OOD detection, utilising the *maximum* logit (commonly referred to as Maxlogit) and energy (i.e., a smooth approximation for the maximum logit), respectively. Moreover, recent studies have incorporated advanced post-processing techniques (Liang et al., 2017; Sun & Li, 2022; Sun et al., 2021) to enhance the performance exhibited by MSP and energy scores. While all these methods focus on extreme information, some studies have adopted a broader perspective, considering information spanned across *ID classes* or *training samples* (Sun et al., 2022; Lee et al., 2018; Ren et al., 2021), which we refer to here as *collective information.* For example, Lee et al. (2018) proposed a method to fit a class-conditional Gaussian distribution on the penultimate layer features based on training samples and derived an OOD score using Mahalanobis distance.

Furthermore, most of these methods, in their respective experimental setups, outperform reported baselines such as MSP and MaxLogit. However, as reported by Zhang et al. (2023b), these methods perform poorly when evaluated within a common benchmarking setup, with some even performing worse than MSP (e.g., ODIN (Liang et al., 2017), GradNorm (Huang et al., 2021), KLM (Hendrycks et al., 2019a)).

In this paper, we introduce a novel approach ExCeL, designed to enhance the consistency in OOD detection by incorporating both extreme and collective information at the output layer. While the logit of the top predicted class (i.e., MaxLogit) (Hendrycks et al., 2019a) captures the extreme information, we show that the probability of other classes in subsequent ranks yields the collective information required to improve the distinguishability of OOD samples. Our approach is motivated by the observation that, during inference, when an input is predicted as a specific ID class, rankings of the remaining classes are more consistently predictable for ID data compared to OOD data. Therefore, each ID class can be characterised by a unique *class rank signature*, that is represented as a *class probability matrix* (CPM) with rows corresponding to predicted ID classes and columns to their ranks. Each element signifies the probability of a particular ID class occurring at a specific rank, computed by analysing predicted class rankings across training samples.

In Figure 1, we show the consistent performance exhibited by ExCeL, compared to existing post-hoc baselines in terms of the aggregate mean overall rank. To summarise, we make the following contributions.

- We show that collective information spanning all classes and training samples, embedded in the output layer of trained DNNs through predicted class ranks, can be used to improve OOD detection. Consequently, we emphasise the existence of a *class rank signature* for each ID class, frequently evident in ID data but not in OOD data.

- We represent the class rank signature as a two-dimensional *class probability matrix* and propose a novel post-hoc OOD detection score named ExCeL, that combines the extreme information provided by the max logit and the collective information provided by the class probability matrix.

- We validate ExCeL through extensive experiments conducted in OpenOOD benchmarking library. Compared to twenty-one existing post-hoc baselines, our method consistently ranks among the top five methods in overall (near and far) OOD detection on CIFAR100, ImageNet-200, and ImageNet-1K datasets. Furthermore, ExCeL exhibits the best performance in terms of the mean overall rank across all datasets outperforming all other baselines – that work best on one dataset but has a performance drop in others.

The rest of the paper is organised as follows. In Section 2, we present the related work, while Section 3 provides the background related to OOD detection. We provide an overview of our methodology in Section 4, followed by a detailed explanation of our experiment setups in Section 5. Next, we present the results of our experiments in Section 6, together with an analysis of the findings and outcomes. In Section 7, we discuss the implications of our findings and the limitations. Finally, Section 8 concludes the paper.

## 2 Related Work

A plethora of recent work attempted to address the challenge of OOD detection (Yang et al., 2021b). These techniques can be broadly categorised into three main groups: *post-hoc inference methods*, *training methods without outlier data*, and *training methods with outlier data* (Zhang et al., 2023b).

### 2.1 Post-hoc inference methods

Post-hoc inference methods (Hendrycks & Gimpel, 2016; Liang et al., 2017; Lee et al., 2018; Liu et al., 2020; Sun et al., 2021; Hendrycks et al., 2019a; Wang et al., 2022; Sun et al., 2022; Sun & Li, 2022; Djurisic et al., 2022; Zhang et al., 2022) utilise post-processors applied to the base classifier. These works formulate an *OOD score*, which, in turn, is employed to produce a binary ID/OOD prediction through thresholding. These methods are active during the inference phase and generally assume that the classifier has been trained using the standard cross-entropy loss. They extract information from either intermediate layers (Sastry & Oore, 2020; Dong et al., 2022; Huang et al., 2021) or the output layer (Hendrycks & Gimpel, 2016; Hendrycks et al., 2019a; Liu et al., 2020; Lee et al., 2018; Ren et al., 2021) of a DNN to establish the OOD score. Given that high-level semantics are more effectively captured in the output layer, much attention has been directed towards exploiting output layer features for OOD detection. Early work by Hendrycks & Gimpel (2016) proposed the maximum softmax probability (MSP) as a reliable baseline for detecting OOD inputs. Building upon this, later studies adopted a similar approach, leveraging more extreme information from the output layer. For instance, subsequent work by Hendrycks et al. (2019a) used the maximum logit directly (i.e., MaxLogit), whereas Liu et al. (2020) employed the energy score, which is a smooth approximation for the maximum logit.

Expanding on these strategies, some studies incorporated advanced post-processing techniques (Liang et al., 2017; Sun & Li, 2022; Sun et al., 2021) to elevate the performance exhibited by MSP and energy scores. For example, Sun et al. (2021) introduced ReAct by rectifying activations at an upper limit, obtaining a modified logit vector with improved OOD separability. While these methods focus on extreme information, others explored a more comprehensive view of the information provided by the output layer (Hendrycks

et al., 2019a; Sun et al., 2022; Lee et al., 2018; Ren et al., 2021). For instance, Hendrycks et al. (2019a)[1] employed KL divergence between the softmax prediction vector and a reference vector to define an OOD score, considering predictions for all *ID classes*. Furthermore, Sun et al. (2022) leveraged information *across training samples*, defining an OOD score based on the distance to the $k^{\text{th}}$ nearest neighbour. However, none of the prior works utilised collective information embedded within the output layer across **both ID classes and training samples**.

## 2.2 Training methods without outlier data

These methods incorporate regularisation techniques during training without relying on auxiliary OOD data, often referred to as outliers (DeVries & Taylor, 2018; Hsu et al., 2020; Tack et al., 2020; Wei et al., 2022; Ming et al., 2022b). They include a diverse range of approaches, such as constraining vector norms (Wei et al., 2022), modifying the decision boundary (Huang & Li, 2021), and applying sophisticated learning methods (Hendrycks et al., 2019b). For instance, Wei et al. (2022) enforced a constant vector norm on the logits to prevent their continuous increase throughout the model training. Furthermore, Huang & Li (2021) proposed to simplify the decision boundary between ID and OOD by decomposing the large semantic space into smaller groups with similar concepts. While the majority of these techniques followed a supervised learning approach, other work (Hendrycks et al., 2019b; Khalid et al., 2022) adopted self-supervised learning for OOD detection.

## 2.3 Training methods with outlier data

In contrast to the methods discussed in Section 2.2, these techniques harness the knowledge derived from auxiliary OOD data during model training (Hendrycks et al., 2018; Yu & Aizawa, 2019; Yang et al., 2021a; Zhang et al., 2023a). This allows OOD detectors to detect OOD inputs more effectively at test time. Within these approaches, some merely incorporate a set of outliers, while others attempt to mine the most informative outliers, a process known as outlier mining (Chen et al., 2021; Ming et al., 2022a). Generally, these methods have the capability to outperform post-hoc and training-based approaches without outlier data, as they expose the model to OOD characteristics to some extent during the training phase. Nonetheless, they have limitations in generalisation since the model gets exposed to only certain types of OODs (**cf.** Section A.3 of the Appendix).

Overall, post-hoc inference methods emerge as the standout choice for OOD detection, owing to their ease of implementation and competitive performance (Yang et al., 2021b). *While existing post-hoc detectors predominantly concentrate on either extreme or collective information, we propose ExCeL that combines both aspects available within the output layer.*

# 3 Background

In classification tasks, the problem of out-of-distribution detection can be defined using the following setup. Let $\mathcal{X} = \mathbb{R}^d$ be the input space and $\mathcal{Y} = \{1, 2, ..., C\}$ be the output space. Assume a deep neural network $f : \mathcal{X} \to \mathbb{R}^{|\mathcal{Y}|}$ is trained on a set of data $D = \{(x_i, y_i)\}_{i=1}^N$ drawn from a distribution $\mathcal{P}$ defined on $\mathcal{X} \times \mathcal{Y}$. The network outputs a logit vector which is used to predict the label of an input sample. Furthermore, let $\mathcal{D}_{\text{in}}$ denote the marginal distribution of $\mathcal{P}$ for $\mathcal{X}$, which represents the distribution of ID data. At test time, the model may encounter inputs from other distributions, denoted as $\mathcal{D}_{\text{out}}$, that differ from $\mathcal{D}_{\text{in}}$, and are recognised as out-of-distribution. Thus, the goal of OOD detection is to define a decision function $g$ to differentiate between ID and OOD inputs. Accordingly, post-hoc detectors leverage a scoring function $S(x)$ and make this decision via a threshold ($\lambda$) comparison as follows.

$$g_\lambda(x) = \begin{cases} \text{in} & \text{if } S(x; f) \geq \lambda \\ \text{out} & \text{if } S(x; f) < \lambda \end{cases} \tag{1}$$

---

[1]This approach is different from the MaxLogit approach in Hendrycks et al. (2019a) – i.e., Hendrycks et al. (2019a) proposed two separate approaches, one relying solely on extreme information (i.e., *MaxLogit*), while the other on collective information (i.e., *KL matching*).



Figure 2: Class rank signatures for top ten ranks for four base ID classes in CIFAR100. Notably, *Bear* often ranks in the top five for *Camel* and *Elephant* classes, while *Bed* frequently appears in the top five for *Chair* and *Table* classes. This is the central concept employed in ExCeL for OOD detection.

## 4 Methodology

In this section, we first explain the intuition behind our proposed method. Following that, we outline the ExCeL score computation algorithm. Lastly, we analytically justify our motivation.

### 4.1 Intuition

As discussed in Section 1, our idea is motivated by the observation that, during inference, when an input is predicted as a specific ID class, the rankings of the subsequent classes are more deterministic for ID data compared to OOD data. To illustrate this more clearly, we depict the probability of a subset of ID classes ranking among the top ten for four base classes of the CIFAR100 dataset in Figure 2. It is important to note that the first rank is always assigned to the respective base class. We notice that when a test input is predicted as either an *elephant* or a *camel*, there is a strong probability of the class *bear* appearing within the top five predictions, given its semantic proximity to these classes. Similarly, in the case of an input predicted as a *chair* or a *table*, the class *bed* is highly likely to be among the top five predictions. We refer to these distinct patterns associated with each class as *class rank signatures*. Notably, such trends are absent in OOD data, enabling us to leverage this information to design ExCeL based on the predicted class ranking for efficient OOD detection.

### 4.2 ExCeL score computation

The ExCeL score computation comprises four steps; two steps involving pre-computation using training samples and two steps executed at test time. Firstly, we calculate the *class probability matrix* (CPM) for each ID class by leveraging correctly classified training samples specific to that class. Following this, the CPM undergoes a smoothing process to amplify the influence of frequently occurring classes while penalising less prevalent ones. During test time, based on the top predicted class of an input, a *rank score* is computed using the relevant CPM. This score captures collective information from the ranking of predicted classes. Finally, the rank score is linearly combined with the max logit to compute the final ExCeL score for OOD detection. These steps are further discussed in the following sections.

#### 4.2.1 Generating the Class Probability Matrix

The objective of the CPM is to model the probability mass function (PMF) across all ID classes within each rank. To achieve this, we start by filtering correctly classified training samples for each specific class, and rank the remaining $C - 1$ classes based on their corresponding logit values. Within each rank, we then calculate the probability of a particular class across the training samples. In Figure 2, when an input is predicted as a *chair*, the probability of class *bed* appearing in one of the next four ranks (i.e., ranks 2-5) is approximately 0.08. Similarly, when an input is predicted as *elephant*, class *bear* appears in ranks 2-4, with a probability of approximately 0.08, in each position.

Likewise, for each ID class $c$, an element $p_{ij}^c$ in the CPM ($P_c \in \mathbb{R}^{C \times C}$) indicates the probability of class $i$ occurring in the $j^{\text{th}}$ rank when an input is predicted as class $c$ as shown in Equation 2.

$$P_c = \begin{pmatrix} p_{11}^c & p_{12}^c & \cdots & p_{1C}^c \\ p_{21}^c & p_{22}^c & \cdots & p_{2C}^c \\ \vdots & \vdots & \ddots & \vdots \\ p_{C1}^c & p_{C2}^c & \cdots & p_{CC}^c \end{pmatrix}, \quad p_{ij}^c = \frac{n_{ij}^c}{N_c} \tag{2}$$

Here, $n_{ij}^c$ represents the number of occurrences where class $i$ appears at rank $j$ among correctly classified samples in class $c$. Furthermore, $N_c$ denotes the total number of correctly classified samples in class $c$, and $C$ represents the total number of ID classes. It is important to highlight that when calculating the probability matrix for any class $c$, we exclusively take into account the correctly classified training samples belonging to that class. Therefore, the top rank in the probability matrix is invariably occupied by the class $c$ itself.

$$p_{i1}^c = \begin{cases} 1 & \text{if } i = c \\ 0 & \text{otherwise} \end{cases}. \tag{3}$$

### 4.2.2 Smoothing the CPM

DNNs tend to exhibit some degree of overfitting to the training set. Therefore, some samples may lose the correlation between similar classes during training. This will induce noise in the CPM computed in Section 4.2.1. Hence, to extract high-level information from the CPM, we incorporate a smoothing step, employing a piecewise function based on the following criteria:

- For classes frequently occurring in a specific rank, a fixed high reward is assigned.

- If the probability of a class, though not highly significant, surpasses that of a random prediction, a small reward is given.

- If the probability is worse than a random prediction but not zero, a small penalty is imposed.

- Classes that do not appear in a specific rank receive a fixed high penalty.

Specifically, let $\hat{P}_c$ be the smoothed probability matrix of $P_c$. Each element $\hat{p}_{ij}^c$ in $\hat{P}_c$ is determined based on the corresponding value $p_{ij}^c$ in $P_c$ based on Equation 4.

$$\hat{p}_{ij}^c = \begin{cases} \frac{a}{C-1} & \text{if } p_{ij}^c \geq \frac{b}{C-1} \\ \frac{1}{C-1} & \text{if } \frac{1}{C-1} \leq p_{ij}^c < \frac{b}{C-1} \\ -\frac{1}{C-1} & \text{if } 0 < p_{ij}^c < \frac{1}{C-1} \\ -\frac{a}{C-1} & \text{if } p_{ij}^c = 0 \end{cases} \tag{4}$$

Here, $a$ corresponds to the reward, while $b$ denotes the high probability threshold. We use the validation set to determine these hyperparameters. The smoothed CPM is then employed to calculate the rank score that provides a measure of how closely a prediction aligns with the distinct class rank signature.

### 4.2.3 Computing the rank score

For a test image $x$, let the predicted class ranking be $[c_1, c_2,..., c_C]$, where $c_1$ and $c_C$ are the classes with the highest and the lowest logit values. Since the top predicted class is $c_1$, we use the smoothed CPM of class $c_1$, denoted by $\hat{P}_{c_1}$ for the rank score calculation. Thus, we compute the rank score ($\mathcal{RS}$) of $x$ as,

$$\mathcal{RS}(x) = \sum_{i=1}^{C} \hat{p}_{c_i i}^{c_1} \tag{5}$$

Since we consider the CPM associated with the top-ranked class when computing the rank score, it is worth noting that the first term (i.e., $\hat{p}_{c_1 1}^{c_1}$) consistently yields 1 (i.e., analogous to $\frac{a}{C-1}$ in $\hat{P}_c$) for all inputs, in accordance with Equation 3. Consequently, the presence of the first term in Equation 5 merely introduces a constant shift to the score, without actively contributing to the discrimination between ID and OOD. However, for completeness, we retain the first term in rank score computation.

Moreover, the rank score can also be computed efficiently via matrix operations. In order to achieve this, the predicted ranking is represented as a one-hot encoded matrix ($\rho$) with the predicted classes as the rows and ranks as columns. For example, in a four-class classification problem, if the predicted class ranking for an input $\hat{x}$ is $[1, 4, 2, 3]$, $\rho_{\hat{x}}$ would be as follows.

$$\rho_{\hat{x}} = \begin{pmatrix} 1 & 0 & 0 & 0 \\ 0 & 0 & 1 & 0 \\ 0 & 0 & 0 & 1 \\ 0 & 1 & 0 & 0 \end{pmatrix} \tag{6}$$

Accordingly, for an input $x$, if the one-hot encoded predicted class ranking matrix is $\rho_x$, we can compute the rank score as,

$$\mathcal{RS}(x) = \text{tr}[(\hat{P}_c)^T \rho_x] \tag{7}$$

A higher rank score is indicative of accumulating more rewards as per Equation 4. This suggests a strong alignment between the predicted class ranking and the rank patterns observed in training samples, indicating that the input is highly likely to be ID. In this way, the rank score encompasses the collective information spanning all ID classes and training samples, which is then utilised to improve OOD detection.

### 4.2.4 Combining with the maximum logit

The rank score draws on collective information from subsequent ranks, excluding the top rank, which itself holds valuable information inherent to ID data. Therefore, as the final step, we combine the rank score with the logit value of the top predicted class, referred to as *MaxLogit* by Hendrycks et al. (2019a), to compute the *ExCeL* score for OOD detection. Since MaxLogit contains extreme information within the output layer, the final *ExCeL* score incorporates both extreme and collective information embedded in the output layer. We define the ExCeL score as a linear combination of the rank score and the MaxLogit as per Equation 8.

$$\text{ExCeL}(x) = \alpha \cdot \mathcal{RS}(x) + (1 - \alpha) \cdot \text{MaxLogit} \tag{8}$$

Here, $\alpha$ balances the use of collective and extreme information for OOD detection. We fine-tune $\alpha$ using a validation set following the same approach used in Section 4.2.2 to fine-tune $a$ and $b$.

### 4.3 Analytical justification

We next analytically justify the existence of distinct patterns in the class ranking that can be exploited to improve OOD detection. Suppose for any predicted ID class, the remaining $C - 1$ classes occur uniformly distributed across the subsequent ranks. Then, for any class $c$, the probability matrix, denoted as Equation 2, takes the following form:

$$p_{ij}^c = \begin{cases} 1 & \text{if } (i = c \text{ and } j = 1) \\ 0 & \text{if } (i \neq c \text{ and } j = 1) \text{ or } (i = c \text{ and } j \neq 1) \, . \\ \frac{1}{C-1} & \text{otherwise} \end{cases} \tag{9}$$

Thus, for any predicted class ranking of an input $x$, the rank score would be,

$$\begin{aligned} \mathcal{RS}(x) &= \sum_{i=1}^{C} \hat{p}_{c_i i}^{c} \\ &= \frac{a}{C-1} + \frac{1}{C-1} + ... + \frac{1}{C-1} \, . \\ &= \frac{a}{C-1} + 1 = k \text{ (constant)} \end{aligned} \tag{10}$$

Subsequently, we can compute the ExCeL score as,

$$
\begin{aligned}
\text{ExCeL}(x) &= \alpha \cdot \mathcal{RS}(x) + (1 - \alpha) \cdot \text{MaxLogit} \\
&= \alpha \cdot k + (1 - \alpha) \cdot \text{MaxLogit}
\end{aligned}
. \tag{11}
$$

As per Equation 11, when classes appear uniformly at random, the rank score remains constant, leading the ExCeL score to correspond to a linearly transformed MaxLogit. In this case, the OOD detection performance of ExCeL would be identical to that of the MaxLogit, since a linear transformation would not impact the separability between ID and OOD. Hence, if ExCeL demonstrates enhanced OOD detection performance compared to MaxLogit, it would affirm the presence of unique class rank signatures within ID classes.

## 5 Experiments

We evaluate ExCeL over common OOD detection benchmarks. To ensure a fair comparison with various baselines, we use the OpenOOD[2] library (Zhang et al., 2023b).

### 5.1 Datasets

We use CIFAR100 (Krizhevsky et al., 2009), ImageNet-200 (a.k.a., TinyImageNet) (Le & Yang, 2015), and ImageNet-1K (Deng et al., 2009) as ID data in our experiments. Each ID dataset is evaluated against near-OOD and far-OOD datasets. As near-OOD data follow a closer distribution to ID compared to far-OOD, near-OOD detection is more challenging than far-OOD detection. For CIFAR100, CIFAR10 and TinyImageNet datasets serve as near-OOD, while MNIST (Deng, 2012), SVHN (Netzer et al., 2011), Textures (Cimpoi et al., 2014), and Places365 (Zhou et al., 2017) are considered as far-OOD. Similarly, for both TinyImageNet and ImageNet-1K, SSB-hard (Vaze et al., 2021) and NINCO (Bitterwolf et al., 2023) datasets are used as near-OOD, while iNaturalist (Van Horn et al., 2018), Textures (Cimpoi et al., 2014), and OpenImage-O (Wang et al., 2022) datasets are used as far-OOD. For consistency, we adopt the same train, validation, and test splits used by the OpenOOD benchmark in implementing our method.

### 5.2 Models

For both CIFAR-100 and TinyImageNet datasets, we use ResNet-18 (He et al., 2016) as the base model. Each model is trained for 100 epochs using the standard cross-entropy loss. We use the SGD optimiser with a momentum of 0.9, a learning rate of 0.1, and a cosine annealing decay schedule (Loshchilov & Hutter, 2016). Furthermore, we incorporate a weight decay of 0.0005, and employ batch sizes of 128 and 256 for CIFAR100 and ImageNet-200, respectively. For ImageNet-1K, we use the pre-trained ResNet-50 (He et al., 2016) model from TorchVision (maintainers & contributors, 2016).

### 5.3 Comparison with baselines

We compare ExCeL with twenty-one existing post-hoc methods provided by the OpenOOD library. These baselines include early OOD detection methods such as maximum softmax probability (MSP) (Hendrycks & Gimpel, 2016), Mahalanobis distance (MDS) (Lee et al., 2018), and OpenMax (Bendale & Boult, 2016), as well as state-of-the-art approaches like SHE (Zhang et al., 2022), ASH (Djurisic et al., 2022), and DICE (Sun & Li, 2022). Furthermore, it is reported that OpenGAN (Kong & Ramanan, 2021) has not shown success on ImageNet-1K and requires substantial changes to make it work with ImageNet-1K models (Zhang et al., 2023b). Therefore, we have not included the results for the OpenGAN baseline for ImageNet-1K.

### 5.4 Evaluation metrics

We employ two metrics to evaluate the OOD detection performance: i) FPR95, which measures the false positive rate of OOD samples when the true positive rate of ID samples is at 95%; ii) AUROC, representing the area under the receiver operating curve. An effective OOD detector will exhibit a low FPR95 alongside

---

[2]https://github.com/Jingkang50/OpenOOD.

Table 1: Comparison of post-hoc OOD detectors for CIFAR100 (ID). Top five values are highlighted, with **ExCeL** and **RMDS** showing the best performance (M.O.R: 1.5), alongside ReAct, GEN, and MaxLogit.

| Post-hoc method | AUROC (%) ↑ | | | FPR95 (%) ↓ | | | M.O.R |
|---|---|---|---|---|---|---|---|
| | Near-OOD | Far-OOD | Overall | Near-OOD | Far-OOD | Overall | |
| OpenMax | 76.41 ± 0.25 (15) | 79.48 ± 0.41 (11) | 77.95 (14) | 56.58 ± 0.73 (9) | 54.50 ± 0.68 (6) | **55.54** (4) | 9.0 |
| MSP | 80.27 ± 0.11 (7) | 77.76 ± 0.44 (14) | 79.02 (12) | **54.80 ± 0.33** (3) | 58.70 ± 1.06 (12) | 56.75 (10) | 11.0 |
| TempScale | **80.90 ± 0.07** (4) | 78.74 ± 0.51 (13) | 79.82 (8) | **54.49 ± 0.48** (2) | 57.94 ± 1.14 (11) | 56.22 (8) | 8.0 |
| ODIN | 79.90 ± 0.11 (10) | 79.28 ± 0.21 (12) | 79.59 (10) | 57.91 ± 0.51 (10) | 58.86 ± 0.79 (13) | 58.39 (13) | 11.5 |
| MDS | 58.69 ± 0.09 (20) | 69.39 ± 1.39 (18) | 64.04 (20) | 83.53 ± 0.60 (19) | 72.26 ± 1.56 (21) | 77.90 (19) | 19.5 |
| MDSEns | 46.31 ± 0.24 (22) | 66.00 ± 0.69 (22) | 56.16 (22) | 95.88 ± 0.04 (22) | 66.74 ± 1.04 (17) | 81.31 (21) | 21.5 |
| RMDS | 80.15 ± 0.11 (9) | **82.92 ± 0.42** (1) | **81.54** (1) | 55.46 ± 0.41 (5) | **52.81 ± 0.63** (3) | **54.14** (2) | **1.5** |
| Gram | 51.66 ± 0.77 (21) | 73.36 ± 1.08 (17) | 62.51 (21) | 92.28 ± 0.29 (21) | 64.44 ± 2.37 (16) | 78.36 (20) | 20.5 |
| EBO | **80.91 ± 0.08** (3) | 79.77 ± 0.61 (8) | 80.34 (7) | 55.62 ± 0.61 (7) | 56.59 ± 1.38 (8) | 56.11 (7) | 7.0 |
| OpenGAN | 65.98 ± 1.26 (18) | 67.88 ± 7.16 (20) | 66.93 (18) | 76.52 ± 2.59 (16) | 70.49 ± 7.38 (19) | 73.51 (16) | 17.0 |
| GradNorm | 70.13 ± 0.47 (17) | 69.14 ± 1.05 (19) | 69.64 (17) | 85.58 ± 0.46 (20) | 83.68 ± 1.92 (22) | 84.63 (22) | 19.5 |
| ReAct | **80.77 ± 0.05** (5) | 80.39 ± 0.49 (6) | **80.58** (4) | 56.39 ± 0.34 (8) | **54.20 ± 1.56** (5) | **55.30** (3) | **3.5** |
| KLM | 76.56 ± 0.25 (14) | 76.24 ± 0.52 (16) | 76.40 (16) | 77.92 ± 1.31 (17) | 71.65 ± 2.01 (20) | 74.79 (17) | 16.5 |
| VIM | 74.98 ± 0.13 (16) | **81.70 ± 0.62** (4) | 78.34 (13) | 62.63 ± 0.27 (14) | **50.74 ± 1.00** (1) | 56.69 (9) | 11.0 |
| KNN | 80.18 ± 0.15 (8) | **82.40 ± 0.17** (2) | **81.29** (3) | 61.22 ± 0.14 (13) | **53.65 ± 0.28** (4) | 57.44 (12) | 7.5 |
| DICE | 79.38 ± 0.23 (11) | 80.01 ± 0.18 (7) | 79.70 (9) | 57.95 ± 0.53 (11) | 56.25 ± 0.60 (7) | 57.10 (11) | 10.0 |
| RankFeat | 61.88 ± 1.28 (19) | 67.10 ± 1.42 (21) | 64.49 (19) | 80.59 ± 1.10 (18) | 69.45 ± 1.01 (18) | 75.02 (18) | 18.5 |
| ASH | 78.20 ± 0.15 (13) | **80.58 ± 0.66** (5) | 79.39 (11) | 65.71 ± 0.24 (15) | 59.20 ± 2.46 (14) | 62.46 (15) | 13.0 |
| SHE | 78.95 ± 0.18 (12) | 76.92 ± 1.16 (15) | 77.94 (15) | 59.07 ± 0.25 (12) | 64.12 ± 2.70 (15) | 61.60 (14) | 14.5 |
| GEN | **81.31 ± 0.08** (1) | 79.68 ± 0.75 (9) | **80.50** (5) | **54.42 ± 0.33** (1) | 56.71 ± 1.59 (9) | **55.57** (5) | **5.0** |
| MaxLogit | **81.05 ± 0.07** (2) | 79.67 ± 0.57 (10) | 80.36 (6) | 55.47 ± 0.66 (6) | 56.73 ± 1.33 (10) | 56.10 (6) | **6.0** |
| ExCeL | 80.70 ± 0.06 (6) | **82.04 ± 0.90** (3) | **81.37** (2) | **55.21 ± 0.56** (4) | **52.24 ± 1.90** (2) | **53.73** (1) | **1.5** |

Table 2: Comparison of post-hoc OOD detectors for ImageNet-200 (ID). Top five values are highlighted, with **ExCeL** and **GEN** showing the best performance (M.O.R: 3.0), alongside KNN, ASH, and TempScale.

| Post-hoc method | AUROC (%) ↑ | | | FPR95 (%) ↓ | | | M.O.R |
|---|---|---|---|---|---|---|---|
| | Near-OOD | Far-OOD | Overall | Near-OOD | Far-OOD | Overall | |
| OpenMax | 80.27 ± 0.10 (13) | 90.20 ± 0.17 (12) | 85.24 (13) | 63.48 ± 0.25 (12) | 33.12 ± 0.66 (8) | 48.30 (12) | 12.5 |
| MSP | **83.34 ± 0.06** (3) | 90.13 ± 0.09 (13) | 86.74 (8) | **54.82 ± 0.35** (2) | 35.43 ± 0.38 (13) | 45.13 (7) | 7.5 |
| TempScale | **83.69 ± 0.04** (1) | 90.82 ± 0.09 (10) | 87.26 (4) | **54.82 ± 0.23** (2) | 34.00 ± 0.37 (9) | 44.41 (6) | **5.0** |
| ODIN | 80.27 ± 0.08 (13) | **91.71 ± 0.19** (5) | 85.99 (11) | 66.76 ± 0.26 (14) | 34.23 ± 1.05 (11) | 50.50 (14) | 12.5 |
| MDS | 61.93 ± 0.51 (19) | 74.72 ± 0.26 (18) | 68.33 (19) | 79.11 ± 0.31 (17) | 61.66 ± 0.27 (17) | 70.39 (17) | 18.0 |
| MDSEns | 54.32 ± 0.24 (22) | 69.27 ± 0.57 (21) | 61.80 (21) | 91.75 ± 0.10 (21) | 80.96 ± 0.38 (20) | 86.36 (21) | 21.0 |
| RMDS | **82.57 ± 0.25** (5) | 88.06 ± 0.34 (16) | 85.32 (12) | **54.02 ± 0.58** (1) | 32.45 ± 0.79 (7) | **43.24** (3) | 7.5 |
| Gram | 67.67 ± 1.07 (18) | 71.19 ± 0.24 (20) | 69.43 (18) | 86.40 ± 1.21 (20) | 84.36 ± 0.78 (21) | 85.38 (20) | 19.0 |
| EBO | 82.50 ± 0.05 (6) | 90.86 ± 0.21 (9) | 86.68 (9) | 60.24 ± 0.57 (9) | 34.86 ± 1.30 (12) | 47.55 (11) | 10.0 |
| OpenGAN | 59.79 ± 3.39 (20) | 73.15 ± 4.07 (19) | 66.47 (20) | 84.15 ± 3.85 (19) | 64.16 ± 9.33 (18) | 74.16 (18) | 19.0 |
| GradNorm | 72.75 ± 0.48 (17) | 84.26 ± 0.87 (17) | 78.51 (17) | 82.67 ± 0.30 (18) | 66.45 ± 0.22 (19) | 74.56 (19) | 18.0 |
| ReAct | 81.87 ± 0.98 (9) | **92.31 ± 0.56** (3) | 87.09 (6) | 62.49 ± 2.19 (11) | **28.50 ± 0.95** (5) | 45.50 (8) | 7.0 |
| KLM | 80.76 ± 0.08 (12) | 88.53 ± 0.11 (15) | 84.65 (16) | 70.26 ± 0.64 (16) | 40.90 ± 1.08 (15) | 55.58 (16) | 16.0 |
| VIM | 78.68 ± 0.24 (16) | 91.26 ± 0.19 (7) | 84.97 (15) | 59.19 ± 0.71 (6) | **27.20 ± 0.30** (1) | **43.20** (2) | 8.5 |
| KNN | 81.57 ± 0.17 (11) | **93.16 ± 0.22** (2) | **87.37** (3) | 60.18 ± 0.52 (8) | **27.27 ± 0.75** (2) | **43.73** (5) | **4.0** |
| DICE | 81.78 ± 0.14 (10) | 90.80 ± 0.31 (11) | 86.29 (10) | 61.88 ± 0.67 (10) | 36.51 ± 1.18 (14) | 49.20 (13) | 11.5 |
| RankFeat | 56.92 ± 1.59 (21) | 38.22 ± 3.85 (22) | 47.57 (22) | 92.06 ± 0.23 (22) | 97.72 ± 0.75 (22) | 94.89 (22) | 22.0 |
| ASH | 82.38 ± 0.19 (8) | **93.90 ± 0.27** (1) | **88.14** (1) | 64.89 ± 0.90 (13) | **27.29 ± 1.12** (3) | 46.09 (9) | **5.0** |
| SHE | 80.18 ± 0.25 (15) | 89.81 ± 0.61 (14) | 85.00 (14) | 66.80 ± 0.74 (15) | 42.17 ± 1.24 (16) | 54.49 (15) | 14.5 |
| GEN | **83.68 ± 0.06** (2) | 91.36 ± 0.10 (6) | **87.52** (2) | **55.20 ± 0.20** (4) | 32.10 ± 0.59 (6) | **43.65** (4) | **3.0** |
| MaxLogit | **82.90 ± 0.04** (4) | 91.11 ± 0.19 (8) | 87.01 (7) | 59.76 ± 0.59 (7) | 34.03 ± 1.21 (10) | 46.90 (10) | 8.5 |
| ExCeL | 82.40 ± 0.04 (7) | **91.97 ± 0.27** (4) | **87.19** (5) | **57.90 ± 0.40** (5) | **28.45 ± 0.80** (4) | **43.18** (1) | **3.0** |

a high AUROC. We also measure the overall performance of each method by computing the mean of near and far-OOD performances. Finally, to compare the performance of a method with other baselines *across ID datasets*, we define the *Mean Overall Rank* (M.O.R) as the average of overall ranks for AUROC and FPR95.

$$\text{Mean Overall Rank} = \frac{\mathcal{R}_{\text{overall}}^{\text{AUROC}} + \mathcal{R}_{\text{overall}}^{\text{FPR95}}}{2} \tag{12}$$

Moreover, we report the mean and the standard deviation of the above metrics computed over three independent runs for CIFAR100 and ImageNet-200, while, for ImageNet-1K, the result for a single run is reported to ensure consistency with the OpenOOD benchmark.

## 5.5 Hyperparameter tuning

Fine-tuning hyperparameters on a validation set is widely adopted in prior OOD detection work (Hendrycks et al., 2018; Sun et al., 2022; Liu et al., 2020). Following a similar approach, we determine the three

parameters associated with ExCeL using the validation set. By performing a grid search on $a$, $b$, and $\alpha$, we discovered the best hyperparameter combination for both CIFAR100 and ImageNet-200 datasets, is $a = 10$, $b = 5$, and $\alpha = 0.8$. For ImageNet-1K, they were $a = 10$, $b = 5$, and $\alpha = 0.6$. We visualise the variation of AUROC for CIFAR100 (ID) in Figure 3.

## 6 Results and Analysis

### 6.1 Comparison with baselines

We present our results in Tables 1, 2, and 3 for CIFAR-100, ImageNet-200, and ImageNet-1K, respectively. Note that due to space limitations, we provide the results on ImageNet-1K (i.e., Table 3) in Section A.1 of the Appendix. Furthermore, we report the average performance for both near-OOD and far-OOD across all OOD benchmarking datasets in each group (**cf.** Section 5.1), along with the overall OOD performance. Per-dataset results can be found in Section A.2. Additionally, we compare ExCeL with outlier-based training methods in Section A.3 of the Appendix.

According to the results, ExCeL consistently stands out among the five top-performing methods in most instances. For instance, in CIFAR100, ExCeL is ranked second in overall AUROC, first in overall FPR95, and has the equal best mean overall rank with RMDS. Similarly, in ImageNet-200, ExCeL is ranked fifth in overall AUROC and first in overall FPR95. Again, ExCeL is ranked equal first in mean overall rank, but this time, sharing it with a different method, GEN. Furthermore, in ImageNet-1K, ExCeL is ranked third in overall AUROC and fifth in overall FPR95, and ranked third in terms of the mean overall rank.

We can also observe from the results that most of the other baselines exhibit strong performance in specific cases but demonstrate only moderate or poor performance in others. For example, RMDS excels in CIFAR100, sharing equal best mean overall rank with ExCeL. However, its performance lags behind in ImageNet-200 and ImageNet-1K. Similarly, GEN performs well in ImageNet-200, but falls behind in CIFAR100. In contrast, ExCeL delivers consistent results in all datasets, achieving a mean overall rank of 1.5 in CIFAR100, 3.0 in ImageNet-200, and 4.0 in ImageNet-1K.

Furthermore, the results also show that ExCeL performs slightly better in far-OOD detection than near-OOD detection. For example, in CIFAR100, ExCeL is ranked second and fourth in terms of FPR95 for far and near OOD, respectively. Similarly, in ImageNet-200, ExCeL is ranked fourth and fifth in terms of FPR95 for far and near OOD. This can be explained using the characteristics of probability matrices in different datasets (**cf.** Section 6.3).

From the three Tables, we can see the variations in the performance of different methods across different evaluation criteria. For example, while ASH performs significantly well on ImageNet-1K (**cf.** Table 3), it falls short on CIFAR-100 (**cf.** Table 1). Similarly, in CIFAR-100 ID scenario (**cf.** Table 1), KNN performs well in terms of AUROC yet falls short with respect to FPR95. Furthermore, on TinyImageNet (**cf.** Table 2), RMDS performs well on near OOD detection but lags behind on far OOD detection in terms of AUROC. Therefore, to combine the performance across ID datasets and measure the consistency across various evaluation criteria, we define *aggregate mean overall rank*, computed as the average of mean overall ranks for different datasets. This metric enables capturing performance insights across all evaluation aspects (i.e., near and far-OOD detection, AUROC and FPR, across three ID datasets). We showed the aggregate mean overall rank of all the methods in Figure 1, and ExCeL clearly stands out for its higher consistency.

### 6.2 The effect of hyperparameters

As mentioned in Section 5.5, we perform the grid search to find $(a,b)$ pair that yields the best performance for the rank score. Next, we show an analysis of the effect of these hyperparameters on the performance using the CIFAR100 dataset as an example. Figure 3a shows the variation of the AUROC on the validation set with $a$ and $b$ when CIFAR100 serves as ID. Notably, the AUROC tends to be particularly low for higher values of both $a$ and $b$. The optimal region for both hyperparameters is found to be $a,b \in \{5,10\}$.

After determining the optimal values for $a$ and $b$, we proceed to fine-tune $\alpha$ to balance between extreme (i.e., MaxLogit) and collective (i.e., rank score) information. Figure 3b illustrates the validation AUROC's

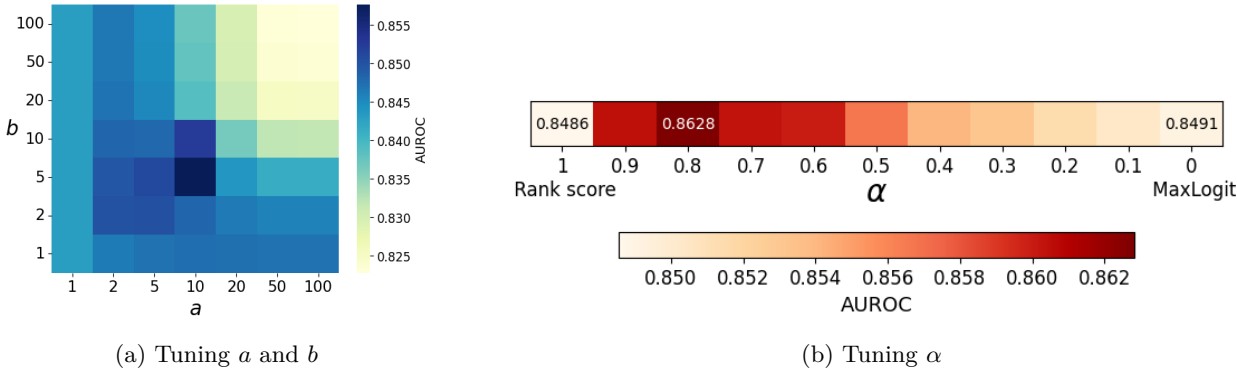

(a) Tuning $a$ and $b$

(b) Tuning $\alpha$

Figure 3: Hyperparameter tuning for CIFAR100 (ID).

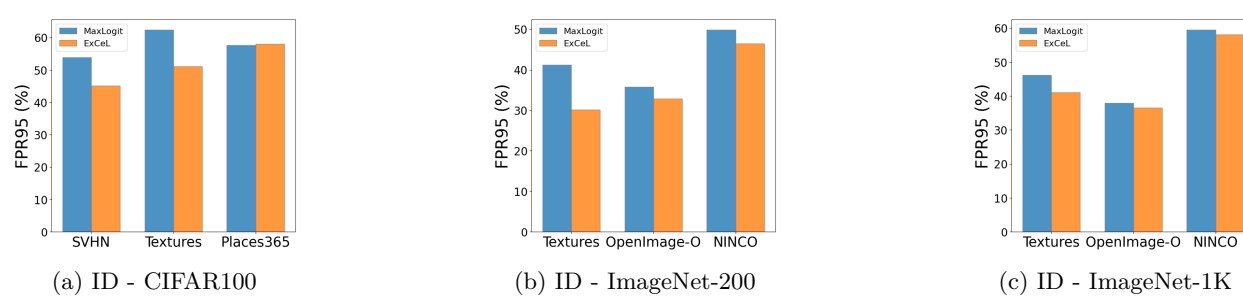

(a) ID - CIFAR100

(b) ID - ImageNet-200

(c) ID - ImageNet-1K

Figure 4: FPR95 comparison between MaxLogit and ExCeL.

variation with respect to $\alpha$. We observe from the extremes in the figure that both the rank score (i.e., $\alpha = 1$) and MaxLogit (i.e., $\alpha = 0$) alone achieve relatively low performance compared to their combinations. For CIFAR100, $\alpha = 0.8$ yields the best performance, as indicated by Figure 3b. We observed similar trends for both ImageNet-200 and ImageNet-1K datasets.

We also note that the performance of the rank score is slightly lower than that of MaxLogit. This is because these classification models are trained on one-hot encoded vectors where the only target is the ground truth class. Therefore, the model is only trained to predict the ground truth class. What we propose from the rank score is discovering some hidden knowledge (i.e., high-level semantic relationships between ID classes) that the model is not specifically trained for and exploiting this knowledge for OOD detection. Therefore, unlike MaxLogit, the rank score itself does not use any information from where the model excels (i.e., leveraging the information from the highest prediction). Therefore, it is possible that the rank score alone would not be able to outperform methods such as MSP, Energy, and MaxLogit that utilise information from the maximum prediction. While it is expected for the rank score to perform slightly lower than the MaxLogit, the performance is not considerably lower than the MaxLogit. In Figure 3b, we see that the performance difference in terms of AUROC between MaxLogit and the rank score is 0.0005. More importantly, the main advantage of the rank score is that it can enhance the performance of MaxLogit without diminishing its original effectiveness, as demonstrated both analytically in Section 4.3 and empirically in Section 6.1.

## 6.3   ExCeL and Maxlogit

As analysed in Section 4.3, an improvement in the OOD detection in ExCeL compared to MaxLogit confirms the existence of unique class rank patterns in ID classes. First, to validate this, we show the FPR95 comparison of ExCeL and MaxLogit in Figure 4 on all datasets. There, we observe two key behaviours. For some OOD datasets, ExCeL achieves a significant improvement in FPR95 compared to MaxLogit, while in others, the performance of ExCeL is on par with MaxLogit. For example, with CIFAR100 as ID, ExCeL significantly improves the OOD detection in SVHN and Textures, while the performance on Places365 is similar to MaxLogit.

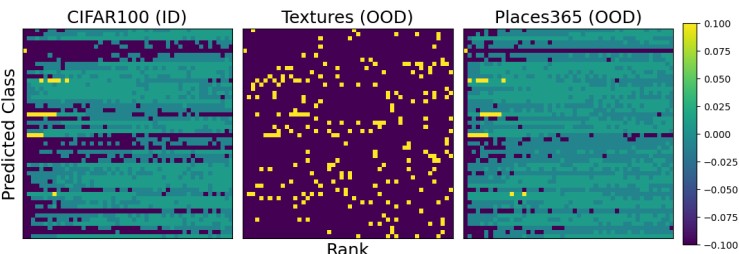

Figure 5: Class probability matrices for ID and OOD samples predicted as a selected ID class in CIFAR100. We can see clear clusters of classes occurring mainly within the top ranks for ID data. In Textures, the predicted class rankings show a random and sparse behaviour, while in Places365, the occurrence of classes is close to a uniformly random distribution.

The reason for this can be explained using the CPMs of ID and OOD data as shown in Figure 5. Here, we show how the subsequent classes are ranked in ID and OOD samples for a selected class in CIFAR100. Specifically, for CIFAR100, we see clear clusters of classes occurring mainly within the top ranks for ID data, indicating a unique class rank signature. In contrast, the class occurrence in Textures data looks random and sparse which allows the ExCeL score to separate the two datasets effectively. On the other hand, the occurrence of classes in Places365 is close to a uniformly random distribution, making the rank score less informative. Consequently, ExCeL performs better against OOD data whose predicted class rankings are sparse and random. In general, this happens more often in far-OOD than near-OOD.

## 7 Discussion

We next discuss the implications of our results and limitations.

**Overall Consistency.** We measure the overall consistency of an OOD detection method using the *aggregate mean overall rank* (**cf.** Section 6.1). This metric embeds the performance information with respect to the two OOD tasks (i.e., near and far-OOD detection), two evaluation metrics (i.e., AUROC and FPR95), and across three ID datasets. A low aggregate mean overall rank with a low standard deviation would indicate consistent OOD detection performance.

Among the compared post-hoc OOD detection methods, ExCeL achieves the lowest aggregate mean overall rank with a low standard deviation, showcasing its consistency across different ID datasets. GEN (Liu et al., 2023) and ReAct (Sun et al., 2021) secure second and third positions in terms of the overall consistency exhibiting competitive performance with ExCeL. Even though ASH (Djurisic et al., 2022) performs exceptionally well in ImageNet-1K, its performance falls short, particularly on the CIFAR100 dataset. Similarly, RMDS (Ren et al., 2021) performs well on CIFAR100, but falls short on ImageNet-1K. Due to their exceptional performance on one particular dataset, both ASH and RMDS achieve a low aggregate mean overall rank. However, the high standard deviation of both ASH and RMDS, as shown in Figure 1, indicates their lack of consistency. This consistency issue was recently highlighted by Zhang et al. (2023b). In this regard, ExCeL's stable results in both near and far OOD detection across multiple datasets are highly promising.

Moreover, we evaluate ExCeL in a common framework compared with many existing baselines since one of the main drawbacks of existing OOD detection methods is that they outperform reported baselines in their own experimental set-ups yet are shown to fail in the common framework (Zhang et al., 2023b). We intend to include ExCeL in the OpenOOD benchmark so that our results can be reproduced and compared with future work.

**Limitations.** ExCeL performs better in scenarios where ID classes consist of a diverse range. For instance, CIFAR100 encompasses a wide range of object categories, including animals (e.g., dogs, cats), vehicles (e.g., cars, aeroplanes), and household items (e.g., chairs, tables), each exhibiting distinct visual features. Similarly, both ImageNet-200 and ImageNet-1K present a broad spectrum of objects, including animals, plants, vehicles, and various other everyday objects. In such situations, we can clearly observe classes in semantic proximity appear in neighbouring ranks, which is exploited by ExCeL. However, when faced with a lack of diversity

among classes, ExCeL's performance may be constrained. Consider CIFAR10, which comprises only ten classes, including six animal classes and four vehicle classes. In this scenario, the distribution of classes within the same category may exhibit high randomness across subsequent ranks. Consequently, the rank signatures are not as prominent as in datasets such as CIFAR100 and ImageNet. Thus, ExCeL's effectiveness is limited under such circumstances.

## 8 Conclusion

In this paper, we proposed a novel OOD score, ExCeL, which combines extreme and collective information within the output layer for consistent OOD detection. We utilised the MaxLogit as extreme information and proposed a novel class rank score that captures information embedded across all ID classes and training samples. We demonstrated that each ID class has a unique signature, determined by the predicted classes in the subsequent ranks, which becomes less pronounced in OOD data. Experiments on CIFAR100, ImageNet-200, and ImageNet-1K showed that ExCeL consistently ranks among the top five methods out of twenty-one baselines. Furthermore, ExCeL showed the equal best performance with RMDS and GEN methods on CIFAR-100 and ImageNet-200 datasets, respectively, in terms of the overall mean rank. With regard to the overall consistency across datasets, ExCeL surpasses all the other post-hoc baselines.

## Acknowledgements

We sincerely thank the Action Editor, Yanwei Fu, and the three anonymous reviewers for their constructive comments. This research was supported by the Australian Government through the Australian Research Council's Discovery Projects funding scheme (Project ID - DP220102520).

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

# A   Appendix

## A.1   Performance on ImageNet-1K

Table 3: Comparison of post-hoc OOD detectors for ImageNet-1K (ID). Top five values are highlighted, with **ASH** showing the best performance (M.O.R: 1.0), alongside ReAct, ExCeL, GEN, and VIM.

| Post-hoc method | AUROC (%) ↑ | | | FPR95 (%) ↓ | | | M.O.R |
|---|---|---|---|---|---|---|---|
| | Near-OOD | Far-OOD | Overall | Near-OOD | Far-OOD | Overall | |
| OpenMax (Bendale & Boult, 2016) | 74.77 (11) | 89.26 (13) | 82.02 (12) | 69.07 (10) | **34.31** (5) | 51.69 (6) | 9.0 |
| MSP (Hendrycks & Gimpel, 2016) | 76.02 (9) | 85.23 (17) | 80.63 (17) | **65.68** (5) | 51.45 (17) | 58.57 (15) | 16.0 |
| TempScale (Guo et al., 2017) | **77.14** (3) | 87.56 (15) | 82.35 (8) | **64.50** (2) | 46.64 (15) | 55.57 (11) | 9.5 |
| ODIN (Liang et al., 2017) | 74.75 (12) | 89.47 (11) | 82.11 (11) | 72.50 (14) | 43.96 (13) | 58.23 (14) | 12.5 |
| MDS (Lee et al., 2018) | 55.44 (19) | 74.25 (19) | 64.85 (19) | 85.45 (18) | 62.92 (18) | 74.19 (18) | 18.5 |
| MDSEns (Lee et al., 2018) | 49.67 (21) | 67.52 (20) | 58.60 (20) | 93.52 (21) | 82.77 (20) | 88.15 (20) | 20.0 |
| RMDS (Ren et al., 2021) | **76.99** (5) | 86.38 (16) | 81.69 (14) | **65.04** (3) | 40.91 (10) | 52.98 (8) | 11.0 |
| Gram (Sastry & Oore, 2020) | 61.70 (18) | 79.71 (18) | 70.71 (18) | 86.63 (19) | 67.36 (19) | 77.00 (19) | 18.5 |
| EBO (Liu et al., 2020) | 75.89 (10) | 89.47 (11) | 82.68 (6) | 68.56 (9) | 38.39 (9) | 53.48 (10) | 8.0 |
| OpenGAN (Kong & Ramanan, 2021)* | N/A | N/A | N/A | N/A | N/A | N/A | N/A |
| GradNorm (Huang et al., 2021) | 72.96 (15) | 90.25 (6) | 81.61 (15) | 78.89 (17) | 47.92 (16) | 63.41 (17) | 16.0 |
| ReAct (Sun et al., 2021) | **77.38** (2) | **93.67** (2) | **85.53** (2) | 66.69 (6) | **26.31** (3) | **46.50** (2) | **2.0** |
| KLM (Hendrycks et al., 2019a) | 76.64 (7) | 87.60 (14) | 82.12 (10) | 72.54 (15) | 46.60 (14) | 59.57 (16) | 13.0 |
| VIM (Wang et al., 2022) | 72.08 (16) | **92.68** (3) | 82.38 (7) | 71.35 (12) | **24.67** (2) | **48.01** (3) | **5.0** |
| KNN (Sun et al., 2022) | 71.10 (17) | 90.18 (7) | 80.64 (16) | 70.87 (11) | **34.13** (4) | 52.50 (7) | 11.5 |
| DICE (Sun & Li, 2022) | 73.07 (14) | **90.95** (4) | 82.01 (13) | 72.43 (13) | 41.83 (12) | 57.13 (12) | 12.5 |
| RankFeat (Song et al., 2022) | 50.99 (20) | 53.93 (21) | 52.46 (21) | 91.83 (20) | 87.17 (21) | 89.50 (21) | 21.0 |
| ASH (Djurisic et al., 2022) | **78.17** (1) | **95.74** (1) | **86.96** (1) | **63.32** (1) | **19.49** (1) | **41.41** (1) | **1.0** |
| SHE (Zhang et al., 2022) | 73.78 (13) | **90.92** (5) | 82.35 (8) | 73.01 (16) | 41.45 (11) | 57.23 (13) | 10.5 |
| GEN (Liu et al., 2023) | 76.85 (6) | 89.76 (8) | **83.31** (4) | **65.32** (4) | 35.61 (6) | **50.47** (4) | **4.0** |
| MaxLogit (Hendrycks et al., 2019a) | 76.46 (8) | 89.57 (10) | **83.02** (5) | 67.82 (8) | 38.22 (8) | 53.02 (9) | 7.0 |
| ExCeL (Ours) | **77.03** (4) | 89.70 (9) | **83.37** (3) | 67.00 (7) | 36.14 (7) | **51.57** (5) | **4.0** |

\* It is reported that OpenGAN has not shown success on ImageNet-1K, and require substantial changes to make it work with ImageNet-1K models (Zhang et al., 2023b).

## A.2   Per-dataset results

This section presents the per-dataset performance for near and far-OOD detection. Table 4 and Table 5 demonstrate the FPR95 and AUROC results, respectively, when CIFAR100 serves as ID. Similarly, Tables 6 and 7 report FPR95 and AUROC results, respectively, with ImageNet-200 as the ID dataset, while Tables 8 and 9 present the same metrics with ImageNet-1K as the ID dataset.

As can be seen from Table 4, in terms of FPR95, ExCeL exhibits the most consistency, ranking among the top five performing methods in five cases (i.e., against ImageNet-200, near-OOD, SVHN, Textures, and far-OOD), which is the highest for any method when CIFAR100 is considered as ID. Similarly, when ImageNet-200 serves as ID, ExCeL ranks among the top five methods in all seven scenarios, as shown in Table 6. Compared to CIFAR100 and ImageNet-200 datasets, ExCeL drops slightly short on the ImageNet-1K dataset, reaching among the top five methods only in two instances, as shown in Table 8. However, ExCeL ranks within the top seven methods in all instances for ImageNet-1K in terms of FPR95.

With regard to AUROC, when CIFAR100 is considered as ID, ExCeL ranks among the top five methods in five cases (i.e., against ImageNet-200, SVHN, Textures, Places365, and Far-OOD), being the most consistent method out of twenty-one baselines, as shown in Table 5. When ImageNet-200 and ImageNet-1K serves as ID, ExCeL ranks among the top five methods in three and two instances, respectively, out of the seven scenarios. Consequently, when the overall consistency is considered, ExCeL outperforms all the other baselines as shown in Figure 1 in terms of the *aggregate mean overall rank* (**cf.** Section 7 in the main text).

Table 4: FPR95 comparison of post-hoc OOD detectors for CIFAR100 (ID). The performance rank of each method is indicated within brackets. Top five values are marked in **bold**.

| Post-processor | CIFAR-10 | ImageNet-200 | Near-OOD | MNIST | SVHN | Textures | Places365 | Far-OOD |
|---|---|---|---|---|---|---|---|---|
| OpenMax | 60.17 ± 0.97 (6) | 52.99 ± 0.51 (10) | 56.58 ± 0.73 (9) | 53.82 ± 4.74 (10) | 53.20 ± 1.78 (10) | 56.12 ± 1.91 (6) | **54.85 ± 1.42** (2) | 54.50 ± 0.68 (6) |
| MSP | **58.91 ± 0.93** (3) | 50.70 ± 0.34 (6) | **54.80 ± 0.33** (3) | 57.23 ± 4.68 (15) | 59.07 ± 2.53 (16) | 61.88 ± 1.28 (10) | 56.62 ± 0.87 (6) | 58.70 ± 1.06 (12) |
| TempScale | **58.72 ± 0.81** (1) | **50.26 ± 0.16** (5) | **54.49 ± 0.48** (2) | 56.05 ± 4.61 (14) | 57.71 ± 2.68 (14) | 61.56 ± 1.43 (9) | **56.46 ± 0.94** (5) | 57.94 ± 1.14 (11) |
| ODIN | 60.64 ± 0.56 (8) | 55.19 ± 0.57 (13) | 57.91 ± 0.51 (10) | **45.94 ± 3.29** (2) | 67.41 ± 3.88 (19) | 62.37 ± 2.96 (12) | 59.71 ± 0.92 (11) | 58.86 ± 0.79 (13) |
| MDS | 88.00 ± 0.49 (20) | 79.05 ± 1.22 (19) | 83.53 ± 0.60 (19) | 71.72 ± 2.94 (19) | 67.21 ± 6.09 (18) | 70.49 ± 2.48 (16) | 79.61 ± 0.34 (18) | 72.26 ± 1.56 (21) |
| MDSEns | 95.94 ± 0.16 (22) | 95.82 ± 0.12 (22) | 95.88 ± 0.04 (22) | **2.83 ± 0.86** (1) | 82.57 ± 2.58 (22) | 84.94 ± 0.83 (20) | 96.61 ± 0.17 (22) | 66.74 ± 1.04 (17) |
| RMDS | 61.37 ± 0.24 (12) | **49.56 ± 0.90** (2) | **55.46 ± 0.41** (5) | 52.05 ± 6.28 (6) | 51.65 ± 3.68 (8) | **53.99 ± 1.06** (4) | **53.57 ± 0.43** (1) | **52.81 ± 0.63** (3) |
| Gram | 92.71 ± 0.64 (21) | 91.85 ± 0.86 (21) | 92.28 ± 0.29 (21) | 53.53 ± 7.45 (9) | **20.06 ± 1.96** (1) | 89.51 ± 2.54 (21) | 94.67 ± 0.60 (21) | 64.44 ± 2.37 (16) |
| EBO | **59.21 ± 0.75** (5) | 52.03 ± 0.50 (9) | 55.62 ± 0.61 (7) | 52.62 ± 3.83 (7) | 53.62 ± 3.14 (11) | 62.35 ± 2.06 (11) | 57.75 ± 0.86 (8) | 56.59 ± 1.38 (8) |
| OpenGAN | 78.83 ± 3.94 (16) | 74.21 ± 1.25 (17) | 76.52 ± 2.59 (16) | 63.09 ± 23.25 (17) | 70.35 ± 2.06 (20) | 74.77 ± 1.78 (18) | 73.75 ± 8.32 (16) | 70.49 ± 7.38 (19) |
| GradNorm | 84.30 ± 0.36 (18) | 86.85 ± 0.62 (20) | 85.58 ± 0.46 (20) | 86.97 ± 1.44 (22) | 69.90 ± 7.94 (22) | 92.51 ± 0.61 (22) | 85.32 ± 0.44 (20) | 83.68 ± 1.92 (22) |
| ReAct | 61.30 ± 0.43 (11) | 51.47 ± 0.47 (7) | 56.39 ± 0.34 (8) | 56.04 ± 5.66 (13) | 50.41 ± 2.02 (7) | **55.04 ± 0.82** (5) | **55.30 ± 0.41** (3) | **54.20 ± 1.56** (5) |
| KLM | 84.77 ± 2.95 (19) | 71.07 ± 0.59 (16) | 77.92 ± 1.31 (17) | 73.09 ± 6.67 (20) | 50.30 ± 7.04 (6) | 81.80 ± 5.80 (19) | 81.40 ± 1.58 (19) | 71.65 ± 2.01 (20) |
| VIM | 70.59 ± 0.43 (14) | 54.66 ± 0.42 (11) | 62.63 ± 0.27 (14) | **48.32 ± 1.07** (3) | **46.22 ± 5.46** (4) | **46.86 ± 2.29** (1) | 61.57 ± 0.77 (13) | **50.74 ± 1.00** (1) |
| KNN | 72.80 ± 0.44 (15) | **49.65 ± 0.37** (3) | 61.22 ± 0.14 (13) | **48.58 ± 4.67** (4) | 51.75 ± 3.12 (9) | **53.56 ± 2.32** (3) | 60.70 ± 1.03 (12) | **53.65 ± 0.28** (4) |
| DICE | 60.98 ± 1.10 (9) | 54.93 ± 0.53 (12) | 57.95 ± 0.53 (11) | **51.79 ± 3.67** (5) | **49.58 ± 3.32** (5) | 64.23 ± 1.65 (14) | 59.39 ± 1.25 (10) | 56.25 ± 0.60 (7) |
| RankFeat | 82.78 ± 1.56 (17) | 78.40 ± 0.95 (18) | 80.59 ± 1.10 (18) | 75.01 ± 5.83 (21) | 58.49 ± 2.30 (15) | 66.87 ± 3.80 (15) | 77.42 ± 1.96 (17) | 69.45 ± 1.01 (18) |
| ASH | 68.06 ± 0.44 (13) | 63.35 ± 0.90 (15) | 65.71 ± 0.24 (15) | 66.58 ± 3.88 (18) | **46.00 ± 2.67** (3) | 61.27 ± 2.74 (8) | 62.95 ± 0.99 (14) | 59.20 ± 2.46 (14) |
| SHE | 60.41 ± 0.51 (7) | 57.74 ± 0.73 (14) | 59.07 ± 0.25 (12) | 58.78 ± 2.70 (16) | 59.15 ± 7.61 (17) | 73.29 ± 3.22 (17) | 65.24 ± 0.98 (15) | 64.12 ± 2.70 (15) |
| GEN | **58.87 ± 0.69** (2) | **49.98 ± 0.05** (4) | **54.42 ± 0.33** (1) | 53.92 ± 5.71 (11) | 55.45 ± 2.76 (13) | 61.23 ± 1.40 (7) | **56.25 ± 1.01** (4) | 56.71 ± 1.59 (9) |
| MaxLogit | **59.11 ± 0.64** (4) | 51.83 ± 0.70 (8) | 55.47 ± 0.66 (6) | 52.95 ± 3.82 (8) | 53.90 ± 3.04 (12) | 62.39 ± 2.13 (13) | 57.68 ± 0.91 (7) | 56.73 ± 1.33 (10) |
| ExCeL (Ours) | 61.07 ± 0.81 (10) | **49.35 ± 0.31** (1) | **55.21 ± 0.56** (4) | 54.67 ± 5.86 (12) | **45.13 ± 0.33** (2) | **51.14 ± 0.14** (2) | 58.02 ± 1.28 (9) | **52.24 ± 1.90** (2) |

Table 5: AUROC comparison of post-hoc OOD detectors for CIFAR100 (ID). The performance rank of each method is indicated within brackets. Top five values are marked in **bold**.

| Post-processor | CIFAR-10 | ImageNet-200 | Near-OOD | MNIST | SVHN | Textures | Places365 | Far-OOD |
|---|---|---|---|---|---|---|---|---|
| OpenMax | 74.38 ± 0.37 (14) | 78.44 ± 0.14 (15) | 76.41 ± 0.25 (15) | 76.01 ± 1.39 (17) | 82.07 ± 1.53 (9) | 80.56 ± 0.09 (6) | 79.29 ± 0.40 (10) | 79.48 ± 0.41 (11) |
| MSP | 78.47 ± 0.07 (6) | 82.07 ± 0.17 (9) | 80.27 ± 0.11 (7) | 76.08 ± 1.86 (16) | 78.42 ± 0.89 (16) | 77.32 ± 0.71 (14) | 79.22 ± 0.29 (11) | 77.76 ± 0.44 (14) |
| TempScale | 79.02 ± 0.06 (4) | 82.79 ± 0.09 (6) | **80.90 ± 0.07** (4) | 77.27 ± 1.85 (13) | 79.79 ± 1.05 (14) | 78.11 ± 0.72 (13) | **79.80 ± 0.25** (5) | 78.74 ± 0.51 (13) |
| ODIN | 78.18 ± 0.14 (7) | 81.63 ± 0.08 (10) | 79.90 ± 0.11 (10) | **83.79 ± 1.31** (2) | 74.54 ± 0.76 (18) | 79.33 ± 1.08 (8) | 79.45 ± 0.26 (8) | 79.28 ± 0.21 (12) |
| MDS | 55.87 ± 0.22 (20) | 61.50 ± 0.28 (20) | 58.69 ± 0.09 (20) | 67.47 ± 0.81 (20) | 70.68 ± 6.40 (20) | 76.26 ± 0.81 (20) | 63.15 ± 0.49 (20) | 69.39 ± 1.39 (18) |
| MDSEns | 43.85 ± 0.31 (22) | 48.78 ± 0.19 (22) | 46.31 ± 0.24 (22) | **98.21 ± 0.78** (1) | 53.76 ± 1.63 (22) | 69.75 ± 1.14 (19) | 42.27 ± 0.73 (22) | 66.00 ± 0.69 (22) |
| RMDS | 77.75 ± 0.19 (11) | 82.55 ± 0.02 (8) | 80.15 ± 0.11 (9) | 79.74 ± 2.49 (7) | **84.89 ± 1.10** (4) | **83.65 ± 0.51** (3) | **83.40 ± 0.46** (1) | **82.92 ± 0.42** (1) |
| Gram | 49.41 ± 0.58 (21) | 53.91 ± 1.58 (21) | 51.66 ± 0.77 (21) | **80.71 ± 4.15** (5) | **95.55 ± 0.60** (1) | 70.79 ± 1.32 (18) | 46.38 ± 1.21 (21) | 73.36 ± 1.08 (17) |
| EBO | **79.05 ± 0.11** (3) | 82.76 ± 0.08 (7) | **80.91 ± 0.08** (3) | 79.18 ± 1.37 (8) | 82.03 ± 1.74 (10) | 78.35 ± 1.32 (11) | 79.52 ± 0.23 (7) | 79.77 ± 0.61 (8) |
| OpenGAN | 63.23 ± 2.44 (18) | 68.74 ± 2.29 (18) | 65.98 ± 1.26 (18) | 68.14 ± 18.78 (19) | 68.40 ± 2.15 (21) | 65.84 ± 3.43 (21) | 69.13 ± 7.08 (18) | 67.88 ± 7.16 (20) |
| GradNorm | 70.32 ± 0.20 (17) | 69.95 ± 0.79 (17) | 70.13 ± 0.47 (17) | 65.35 ± 1.12 (21) | 76.95 ± 4.73 (17) | 64.58 ± 0.13 (22) | 69.69 ± 0.17 (17) | 69.14 ± 1.05 (19) |
| ReAct | **78.65 ± 0.05** (5) | **82.88 ± 0.08** (5) | **80.77 ± 0.05** (5) | 78.37 ± 1.59 (11) | 83.01 ± 0.97 (8) | 80.15 ± 0.46 (7) | **80.03 ± 0.11** (3) | 80.39 ± 0.49 (6) |
| KLM | 73.91 ± 0.25 (15) | 79.22 ± 0.28 (14) | 76.56 ± 0.25 (14) | 74.15 ± 2.59 (18) | 79.34 ± 0.44 (15) | 75.70 ± 0.24 (17) | 75.77 ± 0.37 (16) | 76.24 ± 0.52 (16) |
| VIM | 72.21 ± 0.41 (16) | 77.76 ± 0.16 (16) | 74.98 ± 0.13 (16) | **81.89 ± 1.02** (4) | 83.14 ± 3.71 (7) | **85.91 ± 0.78** (1) | 75.86 ± 0.81 (15) | **81.70 ± 0.62** (4) |
| KNN | 77.02 ± 0.25 (12) | **83.34 ± 0.16** (1) | 80.18 ± 0.15 (8) | **82.36 ± 1.52** (3) | 84.15 ± 1.09 (6) | **83.66 ± 0.83** (2) | 79.43 ± 0.47 (9) | **82.40 ± 0.17** (2) |
| DICE | 78.04 ± 0.32 (10) | 80.72 ± 0.30 (11) | 79.38 ± 0.23 (11) | 79.86 ± 1.89 (6) | **84.22 ± 1.39** (5) | 77.63 ± 0.34 (14) | 78.33 ± 0.66 (13) | 80.01 ± 0.18 (7) |
| RankFeat | 58.04 ± 2.36 (19) | 65.72 ± 0.22 (19) | 61.88 ± 1.28 (19) | 63.03 ± 3.86 (22) | 72.14 ± 1.39 (19) | 69.40 ± 3.08 (20) | 63.82 ± 1.83 (19) | 67.10 ± 1.42 (21) |
| ASH | 76.48 ± 0.30 (13) | 79.92 ± 0.20 (12) | 78.20 ± 0.15 (13) | 77.23 ± 0.46 (14) | **85.60 ± 1.40** (3) | **80.72 ± 0.70** (5) | 78.76 ± 0.16 (12) | **80.58 ± 0.66** (5) |
| SHE | 78.15 ± 0.03 (8) | 79.74 ± 0.36 (13) | 78.95 ± 0.18 (12) | 76.76 ± 1.07 (15) | 80.97 ± 3.98 (13) | 78.74 ± 0.81 (9) | 76.30 ± 0.51 (14) | 76.92 ± 1.16 (15) |
| GEN | **79.38 ± 0.04** (1) | **83.25 ± 0.13** (3) | **81.31 ± 0.08** (1) | 78.29 ± 2.05 (12) | 81.41 ± 1.50 (12) | 78.74 ± 0.84 (10) | **80.28 ± 0.27** (2) | 79.68 ± 0.75 (9) |
| MaxLogit | **79.21 ± 0.10** (2) | **82.90 ± 0.05** (4) | **81.05 ± 0.07** (2) | 78.91 ± 1.47 (10) | 81.65 ± 1.49 (11) | 78.39 ± 0.84 (11) | 79.75 ± 0.24 (6) | 79.67 ± 0.57 (10) |
| ExCeL (Ours) | 78.14 ± 0.09 (9) | **83.26 ± 0.03** (2) | 80.70 ± 0.06 (6) | 78.99 ± 1.73 (9) | **85.91 ± 0.73** (2) | **83.28 ± 0.58** (4) | **79.98 ± 0.57** (4) | **82.04 ± 0.90** (3) |

Table 6: FPR95 comparison of post-hoc OOD detectors for ImageNet-200 (ID). The performance rank of each method is indicated within brackets. Top five values are marked in **bold**.

| Post-processor | SSB-hard | NINCO | Near-OOD | iNaturalist | Textures | OpenImage-O | Far-OOD |
|---|---|---|---|---|---|---|---|
| OpenMax | 72.37 ± 0.11 (12) | 54.59 ± 0.54 (12) | 63.48 ± 0.25 (12) | 24.53 ± 0.96 (8) | 36.80 ± 0.55 (6) | 38.03 ± 0.49 (13) | 33.12 ± 0.66 (8) |
| MSP | **66.00** ± 0.10 (2) | **43.65** ± 0.75 (4) | **54.82** ± 0.35 (2) | 26.48 ± 0.73 (12) | 44.58 ± 0.68 (14) | 35.23 ± 0.18 (9) | 35.43 ± 0.38 (13) |
| TempScale | **66.43** ± 0.26 (3) | **43.21** ± 0.70 (2) | **54.82** ± 0.23 (2) | 24.39 ± 0.79 (6) | 43.57 ± 0.77 (13) | 34.04 ± 0.31 (7) | 34.00 ± 0.37 (9) |
| ODIN | 73.51 ± 0.38 (14) | 60.00 ± 0.80 (14) | 66.76 ± 0.26 (14) | **22.39** ± 1.87 (3) | 42.99 ± 1.56 (12) | 37.30 ± 0.59 (12) | 34.23 ± 1.05 (11) |
| MDS | 83.65 ± 0.47 (18) | 74.57 ± 0.15 (17) | 79.11 ± 0.31 (17) | 58.53 ± 0.75 (17) | 58.16 ± 0.84 (17) | 68.29 ± 0.28 (18) | 61.66 ± 0.27 (17) |
| MDSEns | 92.13 ± 0.05 (22) | 91.36 ± 0.16 (21) | 91.75 ± 0.10 (21) | 83.37 ± 0.70 (20) | 72.27 ± 0.48 (20) | 87.26 ± 0.10 (21) | 80.96 ± 0.38 (20) |
| RMDS | **65.91** ± 0.27 (1) | **42.13** ± 1.04 (1) | **54.02** ± 0.58 (1) | 24.70 ± 0.90 (9) | 37.80 ± 1.32 (7) | 34.85 ± 0.31 (8) | 32.45 ± 0.79 (7) |
| Gram | 85.68 ± 0.85 (19) | 87.13 ± 1.89 (20) | 86.40 ± 1.21 (20) | 85.54 ± 0.40 (21) | 80.87 ± 1.20 (21) | 86.66 ± 1.27 (20) | 84.36 ± 0.78 (21) |
| EBO | 69.77 ± 0.32 (7) | 50.70 ± 0.89 (9) | 60.24 ± 0.57 (9) | 26.41 ± 2.29 (11) | 41.43 ± 1.85 (10) | 36.74 ± 1.14 (11) | 34.86 ± 1.30 (12) |
| OpenGAN | 88.07 ± 2.23 (20) | 80.23 ± 5.71 (18) | 84.15 ± 3.85 (19) | 60.13 ± 9.79 (18) | 66.00 ± 9.97 (18) | 66.34 ± 8.44 (17) | 64.16 ± 9.33 (18) |
| GradNorm | 82.17 ± 0.62 (17) | 83.17 ± 0.21 (19) | 82.67 ± 0.30 (18) | 61.31 ± 2.86 (19) | 66.88 ± 3.59 (19) | 71.16 ± 0.23 (19) | 66.45 ± 0.22 (19) |
| ReAct | 71.51 ± 1.92 (10) | 53.47 ± 2.46 (11) | 62.49 ± 2.19 (11) | **22.97** ± 2.25 (5) | **29.67** ± 1.35 (4) | **32.86** ± 0.74 (2) | **28.50** ± 0.95 (5) |
| KLM | 78.19 ± 2.30 (16) | 62.33 ± 2.66 (16) | 70.26 ± 0.64 (16) | 26.66 ± 1.61 (13) | 50.24 ± 1.26 (16) | 45.81 ± 0.59 (15) | 40.90 ± 1.08 (15) |
| VIM | 71.28 ± 0.49 (9) | 47.10 ± 1.10 (7) | 59.19 ± 0.71 (6) | 27.34 ± 0.38 (14) | **20.39** ± 0.17 (1) | 33.86 ± 0.63 (6) | **27.20** ± 0.30 (1) |
| KNN | 73.71 ± 0.31 (15) | 46.64 ± 0.73 (6) | 60.18 ± 0.52 (8) | 24.46 ± 1.06 (7) | **24.45** ± 0.29 (2) | **32.90** ± 1.12 (3) | **27.27** ± 0.75 (2) |
| DICE | 70.84 ± 0.30 (8) | 52.91 ± 1.20 (10) | 61.88 ± 0.67 (10) | 29.66 ± 2.62 (15) | 40.96 ± 1.87 (8) | 38.91 ± 1.16 (14) | 36.51 ± 1.18 (14) |
| RankFeat | 90.79 ± 0.37 (21) | 93.32 ± 0.11 (22) | 92.06 ± 0.23 (22) | 98.00 ± 0.80 (22) | 99.40 ± 0.68 (22) | 95.77 ± 0.85 (22) | 97.72 ± 0.75 (22) |
| ASH | 72.14 ± 0.97 (11) | 57.63 ± 0.98 (13) | 64.89 ± 0.90 (13) | **22.49** ± 2.24 (4) | **25.65** ± 0.80 (3) | **33.72** ± 0.97 (5) | **27.29** ± 1.12 (3) |
| SHE | 72.64 ± 0.30 (13) | 60.96 ± 1.33 (15) | 66.80 ± 0.74 (15) | 34.38 ± 3.48 (16) | 45.58 ± 2.42 (15) | 46.54 ± 1.34 (16) | 42.17 ± 1.24 (16) |
| GEN | 66.79 ± 0.26 (4) | **43.61** ± 0.61 (3) | **55.20** ± 0.20 (4) | **22.03** ± 0.98 (1) | 42.01 ± 0.92 (11) | **32.25** ± 0.31 (1) | 32.10 ± 0.59 (6) |
| MaxLogit | 69.64 ± 0.37 (6) | 49.87 ± 0.94 (8) | 59.76 ± 0.59 (7) | 25.09 ± 2.04 (10) | 41.25 ± 1.86 (9) | 35.76 ± 0.74 (10) | 34.03 ± 1.21 (10) |
| ExCeL (Ours) | **69.28** ± 0.60 (5) | **46.51** ± 0.20 (5) | **57.90** ± 0.40 (5) | **22.29** ± 1.00 (2) | **30.14** ± 0.64 (5) | **32.91** ± 0.76 (4) | **28.45** ± 0.80 (4) |

Table 7: AUROC comparison of post-hoc OOD detectors for ImageNet-200 (ID). The performance rank of each method is indicated within brackets. Top five values are marked in **bold**.

| Post-processor | SSB-hard | NINCO | Near-OOD | iNaturalist | Textures | OpenImage-O | Far-OOD |
|---|---|---|---|---|---|---|---|
| OpenMax | 77.53 ± 0.08 (13) | 83.01 ± 0.17 (15) | 80.27 ± 0.10 (13) | 92.32 ± 0.32 (11) | 90.21 ± 0.07 (12) | 88.07 ± 0.14 (13) | 90.20 ± 0.17 (12) |
| MSP | **80.38** ± 0.03 (3) | **86.29** ± 0.11 (3) | **83.34** ± 0.06 (3) | 92.80 ± 0.25 (9) | 88.36 ± 0.13 (14) | 89.24 ± 0.02 (9) | 90.13 ± 0.09 (13) |
| TempScale | **80.71** ± 0.02 (2) | **86.67** ± 0.08 (1) | **83.69** ± 0.04 (1) | 93.39 ± 0.25 (7) | 89.24 ± 0.11 (13) | 89.84 ± 0.02 (6) | 90.82 ± 0.09 (10) |
| ODIN | 77.19 ± 0.06 (14) | 83.34 ± 0.12 (13) | 80.27 ± 0.08 (13) | **94.37** ± 0.41 (2) | 90.65 ± 0.20 (8) | **90.11** ± 0.15 (5) | **91.71** ± 0.19 (5) |
| MDS | 58.38 ± 0.58 (20) | 65.48 ± 0.46 (19) | 61.93 ± 0.51 (19) | 75.03 ± 0.76 (19) | 79.25 ± 0.33 (20) | 69.87 ± 0.14 (19) | 74.72 ± 0.26 (18) |
| MDSEns | 50.46 ± 0.36 (22) | 58.18 ± 0.42 (21) | 54.32 ± 0.24 (22) | 62.16 ± 0.73 (21) | 80.70 ± 0.48 (18) | 64.96 ± 0.51 (21) | 69.27 ± 0.57 (21) |
| RMDS | **80.20** ± 0.23 (4) | 84.94 ± 0.28 (9) | **82.57** ± 0.25 (5) | 90.64 ± 0.46 (16) | 86.77 ± 0.38 (15) | 86.77 ± 0.22 (16) | 88.06 ± 0.34 (16) |
| Gram | 65.95 ± 1.08 (18) | 69.40 ± 1.07 (18) | 67.67 ± 1.07 (18) | 65.30 ± 0.20 (20) | 80.53 ± 0.37 (19) | 67.72 ± 0.58 (20) | 71.19 ± 0.24 (20) |
| EBO | 79.83 ± 0.05 (9) | 85.17 ± 0.11 (8) | 82.50 ± 0.05 (6) | 92.55 ± 0.50 (10) | 90.79 ± 0.16 (7) | 89.23 ± 0.26 (10) | 90.86 ± 0.21 (9) |
| OpenGAN | 55.08 ± 1.84 (21) | 64.49 ± 4.98 (20) | 59.79 ± 3.39 (20) | 75.32 ± 3.32 (18) | 70.58 ± 4.66 (21) | 73.54 ± 4.48 (18) | 73.15 ± 4.07 (19) |
| GradNorm | 72.12 ± 0.43 (17) | 73.39 ± 0.63 (17) | 72.75 ± 0.48 (17) | 86.06 ± 1.90 (17) | 86.07 ± 0.36 (17) | 80.66 ± 1.09 (17) | 84.26 ± 0.87 (17) |
| ReAct | 78.97 ± 1.33 (10) | 84.76 ± 0.64 (10) | 81.87 ± 0.98 (9) | 93.65 ± 0.88 (6) | **92.86** ± 0.47 (4) | **90.40** ± 0.35 (2) | **92.31** ± 0.56 (3) |
| KLM | 77.56 ± 0.18 (12) | 83.96 ± 0.12 (12) | 80.76 ± 0.08 (12) | 91.80 ± 0.21 (13) | 86.13 ± 0.12 (16) | 87.66 ± 0.17 (14) | 88.53 ± 0.11 (15) |
| VIM | 74.04 ± 0.31 (16) | 83.32 ± 0.19 (14) | 78.68 ± 0.24 (16) | 90.96 ± 0.36 (15) | **94.61** ± 0.12 (3) | 88.20 ± 0.18 (12) | 91.26 ± 0.19 (7) |
| KNN | 77.03 ± 0.23 (15) | **86.10** ± 0.12 (4) | 81.57 ± 0.17 (11) | **93.99** ± 0.36 (3) | **95.29** ± 0.02 (1) | **90.19** ± 0.32 (3) | **93.16** ± 0.22 (2) |
| DICE | 79.06 ± 0.05 (9) | 84.49 ± 0.24 (11) | 81.78 ± 0.14 (10) | 91.81 ± 0.79 (12) | 91.53 ± 0.21 (6) | 89.06 ± 0.34 (11) | 90.80 ± 0.31 (11) |
| RankFeat | 58.74 ± 0.94 (19) | 55.10 ± 2.52 (22) | 56.92 ± 1.59 (21) | 33.08 ± 4.68 (22) | 29.10 ± 2.57 (22) | 52.48 ± 4.44 (22) | 38.22 ± 3.85 (22) |
| ASH | 79.52 ± 0.37 (7) | 85.24 ± 0.08 (7) | 82.38 ± 0.19 (8) | **95.10** ± 0.47 (1) | **94.77** ± 0.19 (2) | **91.82** ± 0.25 (1) | **93.90** ± 0.27 (1) |
| SHE | 78.30 ± 0.20 (11) | 82.07 ± 0.33 (16) | 80.18 ± 0.25 (15) | 91.43 ± 1.28 (14) | 90.51 ± 0.19 (10) | 87.49 ± 0.70 (15) | 89.81 ± 0.61 (14) |
| GEN | **80.75** ± 0.03 (1) | **86.60** ± 0.08 (2) | **83.68** ± 0.06 (2) | **93.70** ± 0.18 (5) | 90.25 ± 0.10 (11) | **90.13** ± 0.06 (4) | 91.36 ± 0.10 (6) |
| MaxLogit | **80.15** ± 0.01 (5) | **85.65** ± 0.09 (5) | **82.90** ± 0.04 (4) | 93.12 ± 0.45 (8) | 90.60 ± 0.16 (9) | 89.62 ± 0.21 (8) | 91.11 ± 0.19 (8) |
| ExCeL (Ours) | 79.39 ± 0.03 (8) | 85.40 ± 0.04 (6) | 82.40 ± 0.04 (7) | **93.76** ± 0.43 (4) | **92.40** ± 0.05 (5) | 89.75 ± 0.32 (7) | **91.97** ± 0.27 (4) |

Table 8: FPR95 comparison of post-hoc OOD detectors for ImageNet-1K (ID). The performance rank of each method is indicated within brackets. Top five values are marked in **bold**.

| Post-processor | SSB-hard | NINCO | Near-OOD | iNaturalist | Textures | OpenImage-O | Far-OOD |
|---|---|---|---|---|---|---|---|
| OpenMax | 77.33 (10) | 60.81 (12) | 69.07 (10) | **25.29** (3) | 40.26 (6) | 37.39 (6) | **34.31** (5) |
| MSP | **74.49** (3) | 56.88 (6) | **65.68** (5) | 43.34 (17) | 60.87 (19) | 50.13 (15) | 51.45 (17) |
| TempScale | **73.90** (2) | **55.10** (4) | **64.50** (2) | 37.63 (14) | 56.90 (17) | 45.40 (11) | 46.64 (15) |
| ODIN | 76.83 (9) | 68.16 (15) | 72.50 (14) | 35.98 (13) | 49.24 (15) | 46.67 (12) | 43.96 (13) |
| MDS | 92.10 (20) | 78.80 (17) | 85.45 (18) | 73.81 (19) | 42.79 (8) | 72.15 (18) | 62.92 (18) |
| MDSEns | 95.19 (21) | 91.86 (20) | 93.52 (21) | 84.23 (20) | 73.31 (20) | 90.77 (21) | 82.77 (20) |
| RMDS | 77.88 (12) | **52.20** (1) | **65.04** (3) | 33.67 (11) | 48.80 (14) | 40.27 (9) | 40.91 (10) |
| Gram | 89.39 (18) | 83.87 (19) | 86.63 (19) | 67.89 (18) | 58.80 (18) | 75.39 (19) | 67.36 (19) |
| EBO | 76.54 (8) | 60.58 (11) | 68.56 (9) | 31.30 (8) | 45.77 (11) | 38.09 (8) | 38.39 (9) |
| OpenGAN* | N/A | N/A | N/A | N/A | N/A | N/A | N/A |
| GradNorm | 78.24 (14) | 79.54 (18) | 78.89 (17) | 32.03 (9) | 43.27 (9) | 68.46 (17) | 47.92 (16) |
| ReAct | 77.55 (11) | **55.82** (5) | 66.69 (6) | **16.72** (2) | **29.64** (4) | **32.58** (2) | **26.31** (3) |
| KLM | 84.71 (17) | 60.36 (10) | 72.54 (15) | 38.52 (15) | 52.40 (16) | 48.89 (14) | 46.60 (14) |
| VIM | 80.41 (15) | 62.29 (13) | 71.35 (12) | 30.68 (6) | **10.51** (1) | **32.82** (3) | **24.67** (2) |
| KNN | 83.36 (16) | 58.39 (8) | 70.87 (11) | 40.80 (16) | **17.31** (3) | 44.27 (10) | **34.13** (4) |
| DICE | 77.96 (13) | 66.90 (14) | 72.43 (13) | 33.37 (10) | 44.28 (10) | 47.83 (13) | 41.83 (12) |
| RankFeat | 89.63 (19) | 94.03 (21) | 91.83 (20) | 94.40 (21) | 76.84 (21) | 90.26 (20) | 87.17 (21) |
| ASH | **73.66** (1) | **52.97** (2) | **63.32** (1) | **14.04** (1) | **15.26** (2) | **29.15** (1) | **19.49** (1) |
| SHE | 76.30 (7) | 69.72 (16) | 73.01 (16) | 34.06 (12) | **35.27** (5) | 55.02 (16) | 41.45 (11) |
| GEN | **75.73** (4) | **54.90** (3) | **65.32** (4) | **26.10** (4) | 46.22 (13) | **34.50** (4) | 35.61 (6) |
| MaxLogit | 76.20 (6) | 59.44 (9) | 67.82 (8) | **30.61** (5) | 46.17 (12) | 37.88 (7) | 38.22 (8) |
| **ExCeL (Ours)** | **75.82** (5) | 58.18 (7) | 67.00 (7) | 30.85 (7) | 41.07 (7) | **36.51** (5) | 36.14 (7) |

> * It is reported that OpenGAN has not shown success on ImageNet-1K, and require substantial changes to make it work with ImageNet-1K models (Zhang et al., 2023b).

Table 9: AUROC comparison of post-hoc OOD detectors for ImageNet-1K (ID). The performance rank of each method is indicated within brackets. Top five values are marked in **bold**.

| Post-processor | SSB-hard | NINCO | Near-OOD | iNaturalist | Textures | OpenImage-O | Far-OOD |
|---|---|---|---|---|---|---|---|
| OpenMax | 71.37 (13) | 78.17 (13) | 74.77 (11) | 92.05 (7) | 88.10 (13) | 87.62 (10) | 89.26 (13) |
| MSP | 72.09 (6) | 79.95 (9) | 76.02 (9) | 88.41 (15) | 82.43 (19) | 84.86 (16) | 85.23 (17) |
| TempScale | **72.87** (4) | 81.41 (6) | **77.14** (3) | 90.50 (13) | 84.95 (17) | 87.22 (12) | 87.56 (15) |
| ODIN | 71.74 (11) | 77.77 (14) | 74.75 (12) | 91.17 (8) | 89.00 (10) | 88.23 (9) | 89.47 (11) |
| MDS | 48.50 (20) | 62.38 (19) | 55.44 (19) | 63.67 (19) | 89.80 (8) | 69.27 (19) | 74.25 (19) |
| MDSEns | 43.92 (21) | 55.41 (20) | 49.67 (21) | 61.82 (20) | 79.94 (20) | 60.80 (20) | 67.52 (20) |
| RMDS | 71.77 (10) | **82.22** (2) | **76.99** (5) | 87.24 (16) | 86.08 (16) | 85.84 (15) | 86.38 (16) |
| Gram | 57.39 (18) | 66.01 (18) | 61.70 (18) | 76.67 (18) | 88.02 (14) | 74.43 (18) | 79.71 (18) |
| EBO | 72.08 (7) | 79.70 (10) | 75.89 (10) | 90.63 (12) | 88.70 (11) | 89.06 (7) | 89.47 (11) |
| OpenGAN* | N/A | N/A | N/A | N/A | N/A | N/A | N/A |
| GradNorm | 71.90 (9) | 74.02 (17) | 72.96 (15) | **93.89** (3) | 92.05 (6) | 84.82 (17) | 90.25 (6) |
| ReAct | **73.03** (2) | **81.73** (4) | **77.38** (2) | **96.34** (2) | **92.79** (5) | **91.87** (2) | **93.67** (2) |
| KLM | 71.38 (12) | **81.90** (3) | 76.64 (7) | 90.78 (11) | 84.72 (18) | 87.30 (11) | 87.60 (14) |
| VIM | 65.54 (16) | 78.63 (12) | 72.08 (16) | 89.56 (14) | **97.97** (1) | **90.50** (3) | **92.68** (3) |
| KNN | 62.57 (17) | 79.64 (11) | 71.10 (17) | 86.41 (17) | **97.09** (2) | 87.04 (13) | 90.18 (7) |
| DICE | 70.13 (15) | 76.01 (16) | 73.07 (14) | 92.54 (5) | 92.04 (7) | 88.26 (8) | **90.95** (4) |
| RankFeat | 55.89 (19) | 46.08 (21) | 50.99 (20) | 40.06 (21) | 70.90 (21) | 50.83 (21) | 53.93 (21) |
| ASH | **72.89** (3) | **83.45** (1) | **78.17** (1) | **97.07** (1) | **96.90** (3) | **93.26** (1) | **95.74** (1) |
| SHE | 71.08 (14) | 76.49 (15) | 73.78 (13) | 92.65 (4) | **93.60** (4) | 86.52 (14) | **90.92** (5) |
| GEN | 72.01 (8) | **81.70** (5) | 76.85 (6) | 92.44 (6) | 87.59 (15) | **89.26** (4) | 89.76 (8) |
| MaxLogit | **72.51** (5) | 80.41 (8) | 76.46 (7) | 91.17 (8) | 88.39 (12) | **89.17** (5) | 89.57 (10) |
| **ExCeL (Ours)** | **73.26** (1) | 80.80 (7) | **77.03** (4) | 90.79 (10) | 89.24 (9) | 89.07 (6) | 89.70 (9) |

> * It is reported that OpenGAN has not shown success on ImageNet-1K, and require substantial changes to make it work with ImageNet-1K models (Zhang et al., 2023b).

## A.3 Comparison with outlier-based training methods

In Section 2.3, we discussed OOD detection methods that utilise auxiliary outliers during training. On the positive side, they have the potential to outperform other approaches, such as post-hoc methods, due to the additional knowledge provided by outliers. However, they are likely to fall short in standard ID classification accuracy as focusing on outliers during training may lead to a lack of generalisation to the ID data distribution (Zhang et al., 2023b). The model may become specialised in detecting outliers, sacrificing performance on the ID data. Furthermore, they have the tendency to overfit to the seen outliers, which may degrade their performance in OOD detection (Yang et al., 2021b). Accordingly, post-hoc methods are

| Method | FPR95 (%) ↓ | | | ID Test |
| | **Near-OOD** | **Far-OOD** | **Overall** | Accuracy (% ↑) |
|---|---|---|---|---|
| CIFAR-100 | | | | |
| OE (Hendrycks et al., 2018) | **30.73** (1) | 54.82 (3) | **42.78** (1) | 76.84 |
| MCD (Yu & Aizawa, 2019) | 55.88 (4) | 54.39 (2) | 55.14 (3) | 75.83 |
| UDG (Yang et al., 2021a) | 61.42 (5) | 59.00 (4) | 60.21 (5) | 71.54 |
| MixOE (Zhang et al., 2023a) | 55.22 (3) | 63.88 (5) | 59.55 (4) | 75.13 |
| ExCeL (Post-hoc) | 55.21 (2) | **52.24** (1) | 53.73 (2) | **77.25** |
| ImageNet-200 | | | | |
| OE (Hendrycks et al., 2018) | **52.30** (1) | 34.17 (3) | 43.24 (3) | 85.82 |
| MCD (Yu & Aizawa, 2019) | 54.71 (2) | 29.93 (2) | **42.32** (1) | 86.12 |
| UDG (Yang et al., 2021a) | 68.89 (5) | 62.04 (5) | 65.47 (5) | 68.11 |
| MixOE (Zhang et al., 2023a) | 57.97 (4) | 40.93 (4) | 49.45 (4) | 85.71 |
| ExCeL (Post-hoc) | 57.90 (3) | **28.45** (1) | 43.18 (2) | **86.37** |

Table 10: FPR95 comparison of ExCeL with outlier-based training methods. ExCeL, a post-hoc method, leads in far-OOD detection and ranks second overall on both ID datasets.

generally popular due to their competitive performance and ease of implementation. For completeness, in Table 10, we compare ExCeL with four outlier-based training methods: Outlier Exposure (OE) (Hendrycks et al., 2018), Maximum Classifier Discrepancy (MCD) (Yu & Aizawa, 2019), Unsupervised Dual Grouping (UDG) (Yang et al., 2021a), and Mixture Outlier Exposure (MixOE) (Zhang et al., 2023a), to evaluate the aforementioned pros and cons.

From Table 10, we observe that the ID classification accuracy of outlier-based training methods has decreased compared to standard training methods. For example, the test accuracy of UDG has dropped by approximately 6% and 18% on the CIFAR-100 and ImageNet-200 datasets, respectively, compared to standard training, which is used in post-hoc methods. In OOD detection, ExCeL has shown competitive performance with outlier-based methods, achieving the best performance in far-OOD detection and the second-best overall performance on both ID datasets in FPR95.

We observe a significantly higher performance for the OE method in near-OOD detection compared to other methods for the CIFAR-100 dataset. This improvement is attributed to the bias in seen outliers. In the OpenOOD setup, data from *non-overlapping* classes in ImageNet-1K is used as auxiliary outlier data for training OE models on CIFAR-100 and ImageNet-200. Consequently, when TinyImageNet (i.e., the first 200 classes of ImageNet-1K) is evaluated as a near-OOD dataset for the CIFAR-100 ID model, OE performs well because it has already encountered relatively similar OOD inputs during training. This bias of auxiliary outliers during training allows OE to perform significantly better in near-OOD detection for the CIFAR-100 ID scenario. This is further evidenced by OE's performance not being significantly high in other scenarios, even underperforming compared to ExCeL in far-OOD detection.

