# OpenReview forum: "ExCeL: Combined Extreme and Collective Logit Information for Out-of-Distribution Detection"
_TMLR — Accepted by TMLR_

### Review · Reviewer_fTa6 · 2024-10-16

**Summary Of Contributions:**

- The authors identify the existence of a class rank signature for each ID class, frequently evident in ID data but not in OOD data. That is, when a test-time input is predicted as a specific ID class, rankings of the remaining classes are more consistently predictable for ID data compared to OOD data.
- They construct a class probability matrix (CPM) with rows corresponding to predicted ID classes and columns to their ranks, capturing the class rank signature for each ID class. They calculate the matrix based on the training set and smooth the matrix with a proposed transformation.
- The authors propose ExCeL, an OOD detection method by linearly combing the CPM and the MaxLogit, utilizing both collective and extreme information.
- In experiments, ExCeL consistently ranks among the top 5 in overall (near and far) OOD detection on CIFAR100, ImageNet-200, and ImageNet-1K datasets. ExCeL exhibits the best performance in terms of the mean overall rank across all datasets outperforming all other baselines, showing better generalization ability.

**Audience:**

Yes

**Broader Impact Concerns:**

No concersn on the ethical implications.

**Claims And Evidence:**

Yes

**Requested Changes:**

Please see the Strengths And Weaknesses section. If the authors can address my concerns on the Weaknesses of the paper, I would like to recommend acceptance.

**Strengths And Weaknesses:**

## Strengths
- The authors propose an interesting perspective about the rankings of the remaining ID classes.
- Based on OpenOOD library, they have comprehensive experiments on different datasets.
- The proposed method, ExCel, exhibits the best overall performance.

## Weaknesses or Questions
### Major
- There are a few hyper-parameters to tune in ExCel. I’m afraid that the method relies on heavy hyper-parameter tunning in reality. In Section 5.5, the authors mention that they use validation set to tune the hyper-parameters. What validation set do they use? What are the OOD images in the validation set?

- I am not sure whether linear combination of the rank score and the MaxLogit is a contribution of the paper. In principle, we can combine any OOD methods to get possibly better performance. I appreciate the explanation in Section 4.3, but I’m still concerned with it.

(1) What is the performance of the rank score itself for OOD detection?

(2) Is it also possible to combine the rank score with other method that use extreme information, such as MSP or energy?

(3) Previous works, such as ReAct and ASH, also report that their methods can be combined with others to get better performance. If they authors combine the rank score with MaxLogit, I’m not sure whether it is a fair comparison.

- The OpenOOD library has supported newer models like ViT and reports that some methods’ performance drops a lot on different models (for example, ASH on ViT). Could the authors also report the performance of ExCel on the new models, now that they build their study upon OpenOOD as well?


### Minor
- (Page 2) “… with some even performing worse than MSP (e.g., ODIN (Liang et al., 2017), GradNorm (Huang et al., 2021), KLM (Hendrycks et al., 2019a)).”

The citations here can be improved. I feel it’s better to write like “with some even performing worse than MSP, such as ODIN (Liang et al., 2017), GradNorm (Huang et al., 2021), KLM (Hendrycks et al., 2019a)” than to use double parentheses.
- Performance of methods on ImageNet-1k is one of main results for the paper, and is very important because ImageNet-1k is a large-scale dataset for OOD detection evaluation. I think it’s better to consider putting Table 3 in the main manuscript instead of in Appendix.
- I’m curious about the design choice that the authors filter correctly classified training samples for each specific class for building CPM. I feel there are multiple options to explore.

(1) Have the authors tried to use all training samples assigned by the model to the specific class no matter whether their classification is correct?

(2) Or have the authors tried to use training samples with ground truth being that class?

---

> ### Author Response · Authors · 2024-11-25
>
> **Major**
>
> **W1**. **There are a few hyper-parameters to tune in ExCel...**
>
> As mentioned in Section 5.1, we follow the protocol of the OpenOOD benchmark for validation and use the same validation set provided by the OpenOOD library for hyperparameter tuning of existing methods. For the CIFAR-100 in-distribution (ID) scenario, the validation set consists of images from the TinyImageNet dataset. However, these images do not overlap with those used for OOD performance evaluation. Similarly, for the ImageNet ID scenario, the validation set is constructed by OpenOOD from images in the OpenImage-O dataset, ensuring no overlap with the images used for OOD evaluation.
>
> Additionally, having multiple hyperparameters is common for OOD detection methods, including EBO, KNN, ODIN, ASH, DICE, and ReAct, all requiring fine-tuning on a validation set. Even the runner-up method GEN involves tuning parameters Gamma and M, demonstrating that ExCeL's hyperparameter tuning is comparable to existing approaches.
>
> **W2**. **I am not sure whether the linear combination...**
>
> We aim to integrate *extreme* and *collective* information from the output layer, represented by MaxLogit and the rank score, respectively.
>
> Therefore, when looking for possible combinations, we must identify methods representing collective information. Among the methods we tested, KLM and KNN fit into that criterion as they consider information spanned across classes and training samples, respectively.
>
> Below, we demonstrate the performance of combining MaxLogit with these collective methods.
>
> | Method               | FPR95 ↓ |      |        | AUROC ↑ |      |       |
> |----------------------|-------------|------|--------|-------------|------|-------|
> |                      | Near-OOD    | Far-OOD| Overall| Near-OOD    | Far-OOD| Overall|
> | MaxLogit             | 55.47       | 56.73| 56.10  | 81.05       | 79.67| 80.36 |
> | MaxLogit + KLM       | 71.33       | 64.41| 67.87  | 78.01       | 77.40| 77.71 |
> | MaxLogit + KNN       | 58.76       | 53.66| 56.21  | 80.88       | 81.96| 81.42 |
> | ExCeL (Ours)         | **55.21**       | **52.24**| **53.73**  | **81.07**       | **82.04**| **81.56** |
>
>
> ExCeL (MaxLogit + Rank score) has outperformed both KLM and KNN methods when combined with MaxLogit. For instance, compared to the second-best-performing combination (i.e., MaxLogit + KNN), ExCeL achieved a 3.5% improvement in near OOD detection and a 1.5% improvement in far OOD detection in FPR95. While both KNN and KLM notably reduced the performance of the original MaxLogit in at least one task (e.g., KLM in both near and far OOD, and KNN in near OOD detection), ExCeL managed to improve (in far OOD detection) or maintain (in near OOD detection) the performance of MaxLogit without compromise, as shown in Section 4.3.
>
> Within the given time, we could not test other combinations, such as those involving two extreme-based methods (e.g., MSP and MaxLogit). These results can be included in a revised manuscript if required.
>
> **W3**. **What is the performance of the rank score itself...**
>
> The below table shows the individual performance of the rank score on CIFAR-100 and ImageNet-200 datasets.
>
> | ID Dataset    | Near OOD            |                      | Far OOD             |                      |
> |---------------|---------------------|----------------------|---------------------|----------------------|
> |               | AUROC              | FPR95               | AUROC              | FPR95               |
> | CIFAR100      | 77.81 ± 0.28       | 64.12 ± 0.85        | 81.73 ± 0.46       | 53.99 ± 2.13        |
> | ImageNet-200  | 75.95 ± 0.03       | 71.61 ± 0.37        | 87.91 ± 0.60       | 40.48 ± 1.46        |
>
> As discussed in Section 6.2, the rank score performance is lower than that of MaxLogit. However, combining MaxLogit with the rank score significantly improves OOD detection performance compared to either method individually. For example, in the far-OOD detection task on ImageNet-200 (see Table 2), MaxLogit achieves an FPR95 of 34.03, while the rank score achieves 40.48. When combined in ExCeL, the FPR95 improves to 28.45, showing the advantage of integrating the two types of information (i.e., extreme and collective).
>
> **W4**. **Is it also possible to combine the rank score...**
>
> Yes. It is possible to combine the rank score with other extreme metrics, such as MSP or energy, which would result in a behaviour similar to the analysis presented in Section 4.3 due to the non-overlapping type of information (i.e., extreme and collective). However, since we used logit information to construct the rank score, we opted for MaxLogit to maintain consistency.

---

> > ### Author Response · Authors · 2024-11-25
> >
> > **W5**. **Previous works, such as ReAct and ASH,...**
> >
> > We want to clarify that the term "combination" has a different meaning in ReAct/ASH compared to our approach. In ReAct and ASH, the scoring function itself is MaxLogit or Energy, and these methods modify the model’s activation values before applying the scoring function to improve OOD detection. In contrast, our method uses the original model and introduces a scoring function, ExCeL, which combines MaxLogit and RankScore to incorporate both extreme and collective information from the output layer.
> >
> > More specifically, the ReAct paper identifies Energy as the best-performing scoring function based on their experiments. Consequently, in the OpenOOD framework, methods like ReAct, DICE, and ASH are paired with Energy as the default scoring function, as it outperforms MaxLogit in their reported results.
> >
> > Regarding the reviewer’s concern, we believe our comparisons are fair, as methods such as ReAct, DICE, and ASH are inherently paired with scoring functions like Energy, as validated in their original works and the OpenOOD environment.
> >
> > **W6**. **The OpenOOD library has supported newer models like ViT...**
> >
> > We thank the reviewer for this suggestion. Our results are based on the pre-trained models available within the OpenOOD environment. Currently, pre-trained ViT models are unavailable in this environment, so we have not included results for ViT. However, we are happy to provide these results and kindly ask for additional time to complete this evaluation.
> >
> > **Minor**
> >
> > **W7**. **(Page 2) “… with some even performing...**
> >
> > We thank the reviewer for this suggestion. We will revise the manuscript accordingly.
> >
> > **W8**. **Performance of methods on ImageNet-1k...**
> >
> > Due to page limitations, we had to move the results on ImageNet-1K to the Appendix. We are happy to include Table 3 in the main manuscript if additional pages are permitted.
> >
> > **W9**. **I’m curious about the design choice...**
> >
> > We considered both options (1) and (2) during the formulation of ExCeL. However, including misclassified samples introduces noise into the class rankings, as their ranking permutations fail to reflect the true semantic relationships between classes accurately. This results in inconsistencies in the CPM and compromises generalisability. Focusing only on correctly classified training samples ensures that the CPM captures a clean and representative profile for each class, leading to more robust and reliable rankings.

---

### Review · Reviewer_ojac · 2024-10-17

**Summary Of Contributions:**

The paper proposes ExCeL, a method for out-of-distribution (OOD) detection that combines MaxLogit and a novel class rank-based score from a deep neural network's output layer. The method improves OOD detection based on an empirical observation that class rank signatures are more predictable for in-distribution data compared to OOD data. This approach is particularly good because for a uniform distribution of class ranks, it reverts back to an interpretable approach, MaxLogit. Finally, extensive experiments across CIFAR-100, ImageNet-200, and ImageNet-1K demonstrate that ExCeL consistently ranks among the top-performing methods across various benchmarks, outperforming 21 baselines in terms of AUROC and FPR95 metrics.

**Audience:**

Yes

**Claims And Evidence:**

Yes

**Requested Changes:**

- Please address the weaknesses above.

- Can you please update Figure 1 to include the data points in a swarm plot + box plots? I do not see any reason why this should be a bar plot.

- Please perform significance tests on Figure 1 (with multiple testing corrections, ranksum or wilcoxon depending on whether the data is paired) and let us know which algorithms are significantly worse than ExCel. I doubt ExCel will be significantly better than all algorithms as claimed.

- Can you please quantify the class rank signatures discussed in Fig. 2? How do we know that this observation is statistically significant compared to chance level? This example is an helpful illustration, but not a clear evidence.

- To remedy the first weakness, I believe it is important to show one real life application or a realistic simulation (of authors' choosing) beyond just testing on benchmarks. For example, a public brain computer interface dataset or a set of synthetic datasets with controlled OOD data generation processes etc. If the authors simply want to write in the limitation that they have not tested on real life applications, that is also fine, though I strongly encourage them to take the former path.

- Figure 5 suggests that ExCel will simply revert to MaxLogit when the main assumption does not hold, but are there any cases where ExCel would be worse?

**Strengths And Weaknesses:**

## Strengths

- The idea is simple, well presented, and potentially generalizable (though see below) beyond immediate applications. Therefore, this work satisfies the criteria that TMLR's audience would be interested in knowing the findings of this paper.

- Experiments on the benchmark are rigorous, details meticulously provided, and thereby reproducible. If the codebase is also made public, depending on the Action Editor's discretion, this work can receive a reproducibility badge.

- I truly appreciated the discussion of limitations, but I think some of it needs to be extended (see below).


## Weaknesses

- "Our idea is motivated by the observation that, for in-distribution (ID) data, the ranking of classes beyond the predicted class is more deterministic compared to that in OOD data." There are supporting empirical observations and a theoretical motivation in Section 4.3, however, I am not sure how generalizable this observation is. Please correct me if I am wrong, but Section 4.3 simply suggests that if ExCel works, then this motivation is justified. However, there is no apriori prediction as to when ExCel will be successful or not; apart from just testing it. But, in real datasets, we do not know the ground truth, so without a clear theoretical *apriori* motivation, how can we trust in ExCel's generalization capabilities? I think this is one major weakness since generalization is the main claim of this paper (i.e., ExCel works on all benchmarks as opposed to other methods). I will recommend two paths for the authors in the requested changes below to address this.

- Figure 1 claims "ExCeL exhibits the highest consistency across all datasets, indicated by its lowest aggregate mean overall rank." Such claims require statistical tests. Moreover, this figure focuses only on ranks, which is not necessarily interpretable. This needs to be replaced with average accuracies, or this needs to be added as a second figure if the authors like to keep the rank figure. Currently, it makes me think that maybe ExCel was not the one with highest average accuracies.

---

> ### Author Response · Authors · 2024-11-25
>
> **RC1**. **Can you please update Figure 1...**
>
> Please see the below updated figure.
>
> https://drive.google.com/file/d/1qObCZ7I8WZuaISszGDwGh1sA5XSyaP19/view?usp=sharing
>
> We will update the figure in the final manuscript if the paper is accepted for publication.
>
> **RC2**. **Please perform significance tests on Figure 1...**
>
> We performed the *Wilcoxon Signed Rank Test* on the methods and concluded the following.
>
> Null Hypothesis (H0): The median difference in ranks between X and ExCeL is zero, meaning there's no significant difference in performance between the two.
>
> Alternative Hypothesis (H1): The median difference in ranks between X and ExCeL is greater than zero, meaning X has consistently higher (worse) ranks compared to ExCeL.
>
> Methods Significantly Worse than ExCeL: OpenMax, MSP, TempScale, ODIN, MDS, MDSEns, RMDS, Gram, EBO, OpenGAN, GradNorm, KLM, VIM, DICE, RankFeat, SHE, MaxLogit
>
> Methods Not Significantly Worse than ExCeL: ReAct, KNN, ASH, GEN
>
> Our experimental results also demonstrate that these four methods—ReAct, KNN, ASH, and GEN—perform relatively closer to ExCeL compared to other methods.
>
> Please find the details of the significance testing in the link below.
>
> https://drive.google.com/file/d/1rBkv2tW7PlODLwA4GxVEXNjtMSH76Bk6/view?usp=sharing
>
> **RC3**. **Can you please quantify...**
>
> Based on the reviewer's suggestion, we performed *Chi-Square Goodness-of-Fit testing* to evaluate the top five ranks of each displayed class in Figure 2, assessing whether the PMFs shown significantly deviate from chance-level expectations. The hypothesis testing confirmed that these PMFs for the top five ranks are indeed statistically significant compared to chance, supporting the observation of distinct class rank signatures.
>
> Notably, the Chi-square statistic gradually decreases from higher to lower ranks, indicating that beyond a certain rank, the rank patterns may no longer exhibit statistical significance compared to chance levels. Interestingly, the lowest ranks still show patterns that are statistically significant, as they often correspond to classes most distant from the selected class. In contrast, the middle ranks tend to exhibit more statistically insignificant patterns, suggesting less distinct rank signatures in this range.
>
> **RC4**. **To remedy the first weakness,...**
>
> We appreciate the reviewer's thoughtful feedback. To address the concern regarding the generalizability of our observation and provide a theoretical apriori motivation for ExCel, we consider the statistical likelihood of class rank sequences matching the reference rank signature compared to the probabilities associated with other metrics such as MSP or MaxLogit. Specifically, for ID data, the ranking of classes beyond the predicted class exhibits determinism, as discussed in our empirical observations. To illustrate why ranking-based methods like ExCel are theoretically motivated, consider the following:
>
> For a given class in a dataset like CIFAR-10, there are 9 remaining classes, which can occupy the remaining 9 positions in the rank order. This results in 9! (362,880) possible permutations. Thus, the probability of an input's class ranking sequence matching the reference rank signature by chance is 1/9!, which is approximately 2.7×10−6. This probability is exceptionally low. This probability becomes even smaller as the number of classes increases. In contrast, the likelihood of an OOD input exhibiting a high MSP or MaxLogit score is relatively higher. This analysis provides a clear apriori motivation for using ranking information in OOD detection, offering solid theoretical support.

---

> > ### Author Response · Authors · 2024-11-25
> >
> > **RC5**. **Figure 5 suggests that ExCel will simply...**
> >
> > We observed that ExCeL performs worse than MaxLogit (~5% drop in FPR95) in near-OOD detection when CIFAR-10 is considered as the ID dataset.
> >
> > | Method     | Near OOD            |                      | Far OOD             |                      |
> > |------------|---------------------|----------------------|---------------------|----------------------|
> > |            | AUROC              | FPR95               | AUROC              | FPR95               |
> > | MaxLogit   | 87.52 ± 0.47       | 61.32 ± 4.62        | 91.10 ± 0.89       | 41.68 ± 5.27        |
> > | ExCeL      | 86.89 ± 0.23       | 66.55 ± 0.43        | 91.69 ± 0.18       | 40.03 ± 0.84        |
> >
> > As mentioned in the limitations of ExCeL, this drop in performance can be attributed to the lack of diversity in the CIFAR-10 dataset. Unlike datasets such as CIFAR-100 or ImageNet, which encompass a broad spectrum of object categories with distinct semantic and visual features, CIFAR-10 is composed of only ten classes, predominantly consisting of six animal classes and four vehicle classes.
> >
> > This limited diversity results in high randomness in the distribution of classes with similar characteristics (e.g., animals or vehicles) across ranks, making the rank scores less reliable. In such cases, the rank scores may introduce additional variability that affects the performance of MaxLogit, lowering its effectiveness. As a result, ExCeL performs worse than MaxLogit in scenarios like CIFAR-10, where the assumptions about rank-based semantic relationships are less robust due to the lower diversity among classes.

---

> > > ### Comment · Reviewer_ojac · 2024-12-03
> > >
> > > I thank the authors for the revisions. I find the rebuttal convincing and support the publication of this work.

---

### Review · Reviewer_pa32 · 2024-11-11

**Summary Of Contributions:**

This paper addresses the problem of out-of-distribution detection. The approach aims to determine if a certain sample originated from the training distribution or not and have that decision benefit further steps in the automated decision pipeline. Experiments on CIFAR100, ImageNet-200, and ImageNet-1K datasets show that ExCeL is among the top-performing methods in terms of AUROC and FPR95. However, I find that the paper misses a crucial step: the opening sentence of the abstract states 'Deep learning models often exhibit overconfidence in predicting out-of-distribution (OOD) data,' but this assertion was never established. In the context of the datasets used in the experimental section, could this overconfidence actually be shown?

**Audience:**

Yes

**Broader Impact Concerns:**

* The paper addresses a problem of distribution shift that is relevant to many applications. I do not see any broader impact concerns.

**Claims And Evidence:**

Yes

**Requested Changes:**

* From Equation (4), it seems that \hat{p}_{ij}^c can only assume one of four values, was this meant as a scaling factor instead? Or how would one calculate the correlation of a categorically-valued vector?

* The writing frequently uses the phrase 'to be more deterministic' or 'to be less deterministis.' For example, in 'the rankings of classes are more deterministic for ID data compared to OOD data' (section 4.1). What does that mean, 'to be more deterministic'?

* Could Section 3 be expanded and give a few examples of ID and OOD images? E.g. for ImageNet, would one consider an image of a cherry OOD. Or are we talking about spectral properties, e.g. all ImageNet images are encoded with JPEG, so would a PNG image be consider OOD?
Similarly, in Section 5.1, CIFAR10 is considered and OOD class for CIFAR100. How does that work? They were made by the same three authors, who state on their website ‘This dataset [CIFAR100] is just like the CIFAR-10, except it has 100 classes containing 600 images each.’ [https://www.cs.toronto.edu/~kriz/cifar.html].

About the degree of OOD-ness, the paper states that 'the degree of OOD-ness is not a binary concept.' However, the paper does not define what it is. Could the authors provide a definition of OOD-ness, or a reference to a paper that does?
  * A foundational work from 2011 seems relevant to this approach: Torralba, Antonio, and Alexei A. Efros. "Unbiased look at dataset bias." CVPR 2011. IEEE, 2011.
  * Similar to section 5.1, earlier work has already tried to define near-OOD and far-OOD: Djolonga, Josip, et al. "On robustness and transferability of convolutional neural networks." CVPR 2021, IEEE, 2021, c.f. the OOD-ness scale in the ImageNet-C and ImageNet-P datasets.

**Strengths And Weaknesses:**

Strengths:

* Figure 3 clearly shows the considerations for the three most important hyperparameters.

* It is impressive that Table 1 can compare this method to 21 other methods. Also, Table 4, 5, 6 and 7 display a wide range of results, which are useful for future works that want to study this problem.

*  The proposed method can be applied to any pre-trained model, making it versatile and applicable without the need for modifications during the training phase.


Weaknesses:

* The out-of-distribution detection problem is not well defined. Section 3 speaks of ‘marginal probability distributions,’ but any distribution on the image space [0, 255]^{H x W x C} will assign density to any image. The threshold is also not well defined, e.g. ImageNet contains many dogs and no cherries, so an image of a cherry might be OOD, but since computer screens and paintings are in ImageNet, that could be a painting of a cherry.

* The method introduces Class Probability Matrices to compare predictive rank between samples. However, it is not clear where the near-class ranks are learned, is that the test set, train set, or a separate validation set? If it's a separate validation set, how would this method compare to just using that validation data for training? From Figure 3 it seems that the best setting of this method scores only 1 point higher than existing works (at alpha=0), and this gap might well be closed by using the validation set for training.

* Section 5.5 describes how the hyperparameters are different between CIFAR100 or ImageNet. However, there misses an explanation of why these hyperparameters are different. For example, is it coincidental that the a,b hyperparameters are the same, and what can be learnt from the optimal alpha-value being higher for one than the other?

---

> ### Author Response · Authors · 2024-11-25
>
> **W1**. **The out-of-distribution detection problem is not well defined...**
>
> The typical definition of OOD detection involves identifying inputs that are semantically different from the ID classes. For instance, if the ID dataset consists of classes such as cats and dogs, an image of a car would be considered OOD because it belongs to an entirely different semantic category. Marginal probability is commonly used in the literature to define this OOD detection problem [DICE, ReAct, KNN]. For example, if the ID dataset contains classes such as cats and dogs, the marginal probability Pin(x) refers to the probability of x being either a cat or a dog. An OOD image, such as a car, should be further away from the marginal distribution.
>
> The example of a painting of a cherry provided by the reviewer is particularly interesting, as it can be argued to be either ID or OOD, even for humans. While ImageNet includes paintings, it does not have a specific class for cherries, making this case highly ambiguous and an example of a sample that lies on the margin between ID and OOD. This highlights the blurred boundaries inherent in OOD detection and reflects why it remains an unresolved and challenging problem in the field.
>
> **W2**. **The method introduces Class Probability Matrices ...**
>
> As discussed in Section 4.2.1, we are using the training set to learn the class ranks of ID classes.
>
> **W3**. **Section 5.5 describes how the hyperparameters are different...**
>
> The similarity in the a and b hyperparameters across datasets is coincidental. While a=10 and b=5 are the same for these datasets, the actual numerical thresholds, which are computed as a/(C−1) and b/(C−1), respectively, depend on the number of classes C and vary significantly between datasets.
>
> For instance, in CIFAR-100 (C=100), the reward is 0.1, and the high-probability threshold is 0.05. In TinyImageNet (C=200), these values are 0.05 and 0.025, respectively. For ImageNet-1K (C=1000), the reward is 0.01, and the high-probability threshold is 0.005.
>
> The alpha value balances the use of extreme and collective information. A higher alpha places more emphasis on collective information, while a lower alpha prioritises extreme information. We also observed that, for near-OOD detection, relatively lower alpha values (e.g. 0.4) tend to perform better, while for far-OOD detection, higher alpha values (e.g. 0.8) work better. This behaviour is shown in the below example.
>
> |  |      | FPR95 (\%) |      |        |
> |----------|--------------|-------------|------|--------|
> |          |              | Near OOD   | Far OOD | Overall |
> | CIFAR-100| MaxLogit     | 55.5       | 56.7 | 56.1  |
> |          | ExCeL α=0.4| **54.7**       | 55.2 | 54.9  |
> |          |  α=0.8| 55.21       | **52.2** | **53.7**  |
> | MNIST    | MaxLogit     | 21.1       | 2.4  | 11.7  |
> |          | ExCeL α=0.5| 16.5       | **2.3**  | 9.3   |
> |          |  α=0.8| **11.5**       | 3.3  | **7.3**   |

---

> > ### Author Response · Authors · 2024-11-25
> >
> > **RC1**. **From Equation (4), it seems...**
> >
> > The reviewer is correct in observing that \hat{p}_{ij}^c can only assume one of four values. However, this is intended to serve as more of a scaling factor rather than a true categorical variable. These four quantized numerical values are employed to reduce noise in the actual probability estimates. As such, \hat{p}_{ij}^c​ should not be interpreted as a categorical variable.
> >
> > **RC2**. **The writing frequently uses the phrase 'to be more deterministic...**
> >
> > By "more deterministic," we mean "more predictable." For example, in Section 4.1, the phrase "the rankings of classes are more deterministic for ID data compared to OOD data" implies that the rankings for ID data are more consistent and predictable, whereas those for OOD data exhibit greater variability.
> >
> > **RC3**. **Could Section 3 be expanded and give a few examples...**
> >
> > We thank the reviewer for this suggestion. We are happy to expand Section 3 to include examples of ID and OOD images. For instance, in the context of ImageNet, ID images would consist of examples that belong to one of the predefined classes in the dataset, such as a "golden retriever" or a "teapot." On the other hand, OOD images would include those that do not belong to any of the ImageNet classes, such as a "pizza" or a "penguin".
> >
> > As pointed out by the reviewer, the example of a painting of a cherry is an interesting edge case which lies on the margin between ID and OOD. While ImageNet includes paintings, it does not have a specific class for "cherry". This nuance highlights what makes OOD detection more complex and challenging.
> >
> > As for the spectral properties, while the format of an image (e.g., JPEG vs. PNG) might influence its processing, OOD classification typically focuses on the semantic content of the image rather than its technical encoding. Thus, a PNG image could still be considered ID if its content matches one of the defined classes. Conversely, an OOD image would differ in terms of its semantic content rather than its file format.
> >
> > We will add these examples to better clarify the distinction between ID and OOD in Section 3.
> >
> > Regarding CIFAR-10 and CIFAR-100, it’s important to note that these datasets do not have overlapping classes. CIFAR-10 contains 10 distinct classes, while CIFAR-100 contains 100 entirely different classes. Even though the datasets share the same format and were created by the same authors, their class definitions are mutually exclusive. This distinction is why CIFAR-10 is considered OOD for CIFAR-100 in Section 5.1.
> >
> > **RC4**. **About the degree of OOD-ness,...**
> >
> > The "degree of OOD-ness" refers to the extent to which a sample deviates from the distribution of the training data.
> >
> > **Near-OOD**: These are samples that are similar to the ID data but still fall outside the defined classes. For example, if the ID dataset is CIFAR-10, which includes animals like cats and dogs, an image of a fox could be considered near-OOD. While not part of the training classes, it shares visual and semantic similarities with existing categories.
> >
> > **Far-OOD**: These are samples that are entirely unrelated to the ID data in both semantic and visual content. For example, if the ID dataset is CIFAR-10, an image of a T-shirt would be far-OOD. It does not resemble any of the ID categories and falls far from the training distribution.
> >
> > We appreciate the reviewer highlighting these foundational works, including Torralba and Efros (2011) and Djolonga et al. (2021). We will incorporate and cite them in the revised manuscript to provide a clearer context and definition of the degree of OOD-ness.

---

### Decision · Action_Editor_c2Gp · 2025-01-14

**Recommendation:** Accept as is

**Comment:**

The decision to recommend acceptance for the paper ExCeL: Combined Extreme and Collective Logit Information for Out-of-Distribution Detection is based on a combination of its contributions, empirical rigor, theoretical grounding, and responses to reviewer feedback. Below are the key points supporting this decision.


The paper introduces ExCeL, a method that integrates extreme (maximum logit) and collective (rank-based class information) logit features to improve out-of-distribution (OOD) detection. Unlike prior approaches that focus solely on one type of information, this combination provides a more consistent and robust framework. This novel dual-feature perspective is well-motivated and represents a significant contribution to the OOD detection domain.

The theoretical reasoning behind using rank-based class signatures is well-justified. The authors show that deterministic ranking patterns emerge in ID data, making this feature useful for distinguishing OOD samples.

The authors have thoroughly responded to reviewer comments, improving clarity and addressing specific weaknesses, and all reviewers accept this paper:
They provide expanded explanations of OOD detection challenges, include examples of ID vs. OOD cases, and discuss the limitations of ExCeL in small datasets like CIFAR-10.
Statistical significance tests and improved visualization (swarm plots and box plots) enhance the robustness of their conclusions.
The combination with MaxLogit was justified with empirical results showing that ExCeL outperforms similar combinations involving collective methods (e.g., KLM and KNN).

**Audience:**

This paper is interested to Machine Learning and Deep Learning Researchers, and Statistical and Theoretical AI Researchers.

**Claims And Evidence:**

The claims made in the submission are effectively supported by clear, accurate, and convincing evidence. The authors of ExCeL: Combined Extreme and Collective Logit Information for Out-of-Distribution Detection provide a well-structured argument linking the proposed method to its performance improvements:

The authors propose ExCeL, a novel approach that integrates both extreme (maximum logit) and collective (rank-based class information) features for out-of-distribution (OOD) detection. They claim that this dual use of information enhances performance consistency across datasets.

Evidence Supporting the Claims: Extensive empirical results demonstrate that ExCeL consistently ranks among the top five performers across 21 post-hoc OOD detection baselines on multiple datasets, including CIFAR-100, ImageNet-200, and ImageNet-1K. Performance is quantified using standard metrics like AUROC and FPR95, with detailed tables and figures comparing ExCeL against other methods​.

Statistical tests  provide significant evidence that ExCeL outperforms many baseline methods. These tests validate the robustness of the ranking-based approach by showing that most compared methods exhibit statistically worse performance​.

Clear theoretical motivation is presented, highlighting why rank-based signatures provide more deterministic behavior for in-distribution (ID) data than for OOD data. Probability calculations for class rank permutations further reinforce the method’s soundness​.

Rebuttals Addressing Reviewer Concerns: Authors clarify the generalization capabilities by providing theoretical probabilities demonstrating how the rank structure emerges from ID data distributions. They also acknowledge specific limitations, such as reduced performance on small, less diverse datasets like CIFAR-10, and provide empirical evidence explaining these scenarios​.

The combination of rank score and MaxLogit is explained as a deliberate choice to balance extreme and collective information, with results showing superior performance compared to using these metrics alone or with other combinations​.

Overall, the combination of strong empirical validation, statistical rigor, and thoughtful rebuttals ensures that the claims are well-supported by compelling evidence throughout the submission.